

# Archimedean screw in driven chiral magnets

**Nina del Ser⋆, Lukas Heinen and Achim Rosch**

Institute for Theoretical Physics, University of Cologne, 50937 Cologne, Germany

⋆ nd374@cantab.ac.uk

The supplementary video files can be found at https://arxiv.org/src/2012.11548v4/anc.

## Abstract

In chiral magnets a magnetic helix forms where the magnetization winds around a propagation vector q. We show theoretically that a magnetic field $B_\perp(t) \perp q$, which is spatially homogeneous but oscillating in time, induces a net rotation of the texture around q. This rotation is reminiscent of the motion of an Archimedean screw and is equivalent to a translation with velocity $v_{\text{screw}}$ parallel to q. Due to the coupling to a Goldstone mode, this non-linear effect arises for arbitrarily weak $B_\perp(t)$ with $v_{\text{screw}} \propto |B_\perp|^2$ as long as pinning by disorder is absent. The effect is resonantly enhanced when internal modes of the helix are excited and the sign of $v_{\text{screw}}$ can be controlled either by changing the frequency or the polarization of $B_\perp(t)$. The Archimedean screw can be used to transport spin and charge and thus the screwing motion is predicted to induce a voltage parallel to q. Using a combination of numerics and Floquet spin wave theory, we show that the helix becomes unstable upon increasing $B_\perp$, forming a 'time quasicrystal' which oscillates in space and time for moderately strong drive.

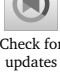

# 1  Introduction

The Archimedean screw has benefited humanity as a mechanical tool since antiquity.

There is evidence that it was already used in ancient Egypt to pump water, but even to this day, it is still used extensively, for example to transport materials such as powders and grains in factories. In addition, some bacteria use helical screws, so-called flagella, to propel themselves through liquids. Usually an Archimedean screw consists of a helical surface encased in a tilted tube, a simplified version of which is shown in Fig. 1(a). By rotating the screw on its axis as shown, the helical surface can be made to push material inside upwards, as indicated by the blue spheres and vertical arrows.

Helical surfaces analogous to the Archimedean screw have been predicted and observed in chiral magnets [1–3]. There the helical surface is spanned be spins winding around the corresponding pitch vector **q**, see Fig. 1(b). These structures form naturally in chiral magnets at low temperatures [4, 5]. Chiral magnets are dominantly ferromagnetic materials in which inversion symmetry is broken by the crystal lattice, allowing weak spin–orbit interactions to induce a so-called Dzyaloshinskii-Moriya interaction. It is the competition between these two interactions that favors the formation of long-wavelength helical structures [3]. In addition to the helical phase, chiral magnets also host other phases. The conical phase can simply be viewed as a helical phase oriented parallel to an external magnetic field where spins uniformly tilt towards the magnetic field. In a small phase pocket close to the critical temperature $T_c$, a skyrmion phase — a lattice of topologically quantized magnetic whirls — can form. Skyrmion phases can be manipulated by ultrasmall external forces created, e.g., by electric currents [6,7]. The coupling to currents is directly proportional to the winding number of skyrmions. This mechanism is absent for the topologically trivial helical and conical phases, which are therefore more difficult to control. It has also been suggested to use oscillating fields to move a single skyrmion [8–10], to create skyrmions [11] or to melt skyrmion crystals [12]. Similarly, the motion of domain walls by oscillating fields has been studied in simulations [13].

When a weak oscillating magnetic field is applied to a magnet, to linear order in perturbation theory, spin waves are excited. Early experiments by Onose et al. [14] showed that the helical, conical, skyrmion lattice and ferromagnetic phases exhibit a characteristic pattern of collective spin wave excitations. These excitations have been quantitatively described by

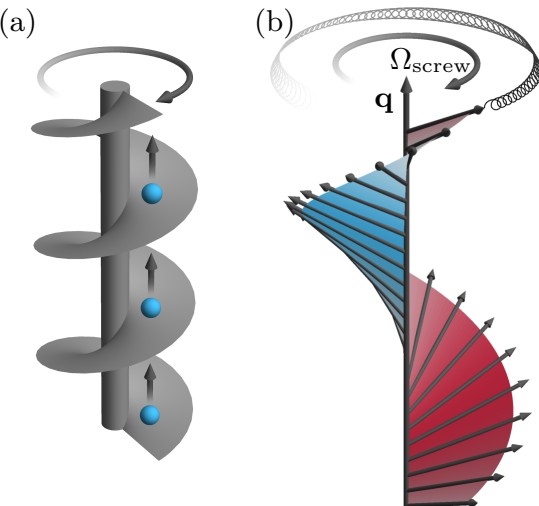

Figure 1: Panel (a) Simplified illustration of an Archimedean screw. A rotation of the screw induces an upwards motion of the material inside (blue spheres).
Panel (b) Dynamics of a conical state driven by an oscillating magnetic field perpendicular to **q**. In fading black the oscillations of a selected spin is shown at the top of the figure. The oscillating field induces a fast precession which triggers a slow screw motion of the magnetic texture. An animated version of this figure is shown as a supplementary video

linear spin wave theory in a range of different materials [15–18]. In the case of the helical and conical states at $\mathbf{k} = 0$, the oscillating external field couples to two modes, often referred to as $\pm Q$ modes, for a review see [17]. They can be viewed as (spin-compression) waves traveling up or down the helix.

To second order in perturbation theory, a magnetic field oscillating with the frequency $\Omega$ is expected to generate a response at frequencies 0 and $2\Omega$. We will argue that the zero-frequency response couples to the Goldstone modes of the helical and conical phases. Here, naïve perturbation theory breaks down and a slow precessional motion with frequency $\Omega_{\text{screw}}$ of all spins is induced as sketched in Fig. 1(b). This type of motion is precisely of the type characteristic of an Archimedean screw. Equivalently, the net rotation can also be interpreted as a translation with velocity $V_{\text{screw}} = \lambda \Omega_{\text{screw}}/(2\pi)$, where $\lambda$ is the pitch of the helix. In the absence of pinning by disorder, this screw-like motion is induced for arbitrarily weak oscillating fields.

Upon increasing the strength of the driving field, the Archimedean screw solution ultimately becomes unstable. The discrete time-translational invariance of the driven system is spontaneously broken and an incommensurate spin wave oscillating in space and time is macroscopically occupied. Such a state can be viewed as a time crystal, or, more precisely, as a time quasicrystal as it is an incommensurate state [19, 20]. In magnets such states are also referred to as magnon Bose-Einstein condensates (BECs). Such magnon BECs have, for example, been observed in YIG samples driven by GHz frequencies [21, 22].

In the following, we will first analyze the equations of motion of spins in a helical or conical state driven by a perpendicular magnetic field $\mathbf{B}_1(t)$ to second order in the amplitude $\mathcal{O}(B_1^2)$. We will show that a screw-like motion is induced and compare our analytical results to numerical micromagnetic simulations. To investigate the stability of the perturbative solution, we calculate the Floquet spin wave spectrum of the system and identify leading instabilities. Micromagnetic simulations show that these instabilities lead to the formation of a time qua-

sicrystal at intermediate driving strength while chaotic behavior sets in at stronger driving. Finally, we show how the helical or conical state can be used as an Archimedean screw to transport electrons.

## 2 Model

We consider a chiral magnet in the presence of Dzyaloshinskii-Moriya interactions described by the free energy

$$F = \int d^3 r \left[ -\frac{J}{2} \hat{\mathbf{M}} \cdot \nabla^2 \hat{\mathbf{M}} + D \hat{\mathbf{M}} \cdot (\boldsymbol{\nabla} \times \hat{\mathbf{M}}) - \mathbf{M} \cdot \mathbf{B}_{\text{ext}} \right] + F_{\text{demag}}[\mathbf{M}], \tag{1}$$

where $F_{\text{demag}}[\mathbf{M}]$ encodes the dipole-dipole interactions and we use Heisenberg spins of fixed length, $|\mathbf{M}| = M_0$ with $\hat{\mathbf{M}} = \mathbf{M}/M_0$. We consider an external magnetic field

$$\mathbf{B}_{\text{ext}} = \mathbf{B}_0 + \epsilon \mathbf{B}_1, \qquad \mathbf{B}_0 = (0, 0, B_0)^T, \qquad \mathbf{B}_1(t) = (B_\perp^x \cos(\Omega t), B_\perp^y \sin(\Omega t), 0)^T. \tag{2}$$

We will consider both linearly polarized fields, $B_\perp^y = 0$, and circular polarization, $B_\perp^x = \pm B_\perp^y$. Throughout the paper we consider small oscillating fields and use $\epsilon \ll 1$ for bookkeeping purposes in perturbation theory.

We have performed all analytical and numerical calculations in the presence of dipolar interactions. To avoid overly long formulas, the analytical formulas presented in the main text are given in the *absence* of dipolar interactions. The effects of dipolar interactions are discussed in App. C.

In the absence of oscillating fields, $\epsilon = 0$, the free energy for $B_0 < \frac{M_0 D^2}{J}$ is minimized by the conical state described by

$$\mathbf{M} = M_0 \left( \sin(\theta_0) \cos(qz), \quad \sin(\theta_0) \sin(qz), \quad \cos(\theta_0) \right)^T, \tag{3}$$

where the helical pitch vector and the conical angle are given by $\mathbf{q} = \frac{D}{J} \hat{\mathbf{z}}$ and $\cos(\theta_0) = \frac{B_0 M_0 J}{D^2}$, respectively. When $B_0 = 0$ the free energy is isotropic and there is no preferred direction for $\mathbf{q}$, but we are still free to choose the $z$-axis as the direction of spontaneous symmetry breaking, $\mathbf{q} \parallel \mathbf{z}$. In this case $\theta_0 = \pi/2$, corresponding to a helical state where magnetization and $\mathbf{q}$ are perpendicular to each other everywhere.

The magnetic texture, Eq. (3), is translationally invariant in the $xy$ plane. It is also invariant under a combined spin-rotation and translation along the $\hat{z}$ direction.

## 3 Archimedean screw

For an oscillating field, we calculate the time evolution of $\mathbf{M}(\mathbf{r}, t)$ using the Landau-Lifshitz-Gilbert (LLG) equation

$$\dot{\mathbf{M}} = \gamma \mathbf{M} \times \mathbf{B}_{\text{eff}} - \frac{\gamma}{|\gamma|} \alpha \hat{\mathbf{M}} \times \dot{\mathbf{M}}. \tag{4}$$

Here $\mathbf{B}_{\text{eff}} = -\frac{\delta F[M]}{\delta \mathbf{M}}$ is functional derivative of the free energy Eq. (1), $\alpha$ is a phenomenological damping term and $\gamma$ is the gyromagnetic ratio. Note that we use a convention where $\gamma$ is negative, with $\gamma = -\frac{|e|g}{2m_e}$ for an electron with charge $-|e|$, mass $m_e$ and $g$-factor $g$. The prefactor in front of the damping term ensures that all formulas remain valid independent of the sign of $\gamma$.

The goal is to calculate the response to the oscillating magnetic field, Eq. (2), by doing a Taylor expansion in $\epsilon$. To this end we now update the parametrisation of $\mathbf{M}$ given in Eq. (3) to allow for some small dynamical excitations, replacing

$$
\begin{aligned}
\theta_0 &\to \theta_0 + \epsilon\theta_1(z,t) + \epsilon^2\theta_2(z,t) + O(\epsilon^3), \\
qz &\to qz + \epsilon\phi_1(z,t) + \epsilon^2\phi_2(z,t) + O(\epsilon^3).
\end{aligned}
\tag{5}
$$

This parametrization assumes that the system remains translationally invariant in the $xy$ plane. Importantly, we assume here that the $\mathbf{q}$ vector does not tilt in the presence of the oscillating magnetic field. This is justified as such a tilt would nominally lead to frictional forces diverging in the thermodynamic limit. Experimentally, such a tilting occurs only for extremely slow changes of the field direction [23].

Substituting Eq. (5) into Eq. (4) and dotting with $\frac{\partial\mathbf{M}}{\partial\theta}, \frac{\partial\mathbf{M}}{\partial\phi}$ gives two sets of coupled differential equations for $\theta_{1,2}, \phi_{1,2}$. To first order we get

$$
\begin{aligned}
\mathrm{sgn}(\gamma)\dot\theta_1 - \alpha s\dot\phi_1 &= -s\phi_1'' + b_x(t)\sin(z) - b_y(t)\cos(z), \\
\mathrm{sgn}(\gamma)s\dot\phi_1 + \alpha\dot\theta_1 &= \theta_1'' - s^2\theta_1 + cb_x(t)\cos(z) + cb_y(t)\sin(z).
\end{aligned}
\tag{6}
$$

The equations to second order in $\epsilon$ take the form

$$
\begin{aligned}
\mathrm{sgn}(\gamma)\dot\theta_2 - \alpha(s\dot\phi_2 + c\theta_1\dot\phi_1) =&\; -2c\theta_1'\phi_1' - c\theta_1\phi_1'' - s\phi_2'' \\
&+ \phi_1\big(b_x(t)\cos(z) + b_y(t)\sin(z)\big), \\
\mathrm{sgn}(\gamma)s(2c\theta_1\dot\phi_1 + s\dot\phi_2) + \alpha(c\theta_1\dot\theta_1 + s\dot\theta_2) =&\; s\theta_2'' + c\theta_1\theta_1'' - s^2 c\phi_1'^2 - \frac{5}{2}cs^2\theta_1^2 - s^3\theta_2 \\
&+ (c^2 - s^2)\theta_1[b_x(t)\cos(z) + b_y(t)\sin(z)] \\
&+ sc\phi_1\big[b_y(t)\cos(z) + b_x(t)\sin(z)\big],
\end{aligned}
\tag{7}
$$

with $c = \cos(\theta_0)$ and $s = \sin(\theta_0)$. Note that we switched to dimensionless units $b_{x,y} = \frac{B_{x,y}M_0 J}{D^2}$, where $b_x(t) = b_x\cos(\omega t), b_y(t) = b_y\sin(\omega t)$ and also use dimensionless space and time units: $z \to q^{-1}z$ and $t \to \frac{JM_0}{D^2|\gamma|}t$, respectively. The latter also motivates the definition of a dimensionless driving frequency $\omega = \frac{JM_0}{D^2|\gamma|}\Omega$. Eq. (6) and (7) are to be solved consecutively, as the first order solutions $\theta_1, \phi_1$ enter in the second order equations.

To linear order, $\mathcal{O}(\epsilon^1)$, the driving terms proportional to $b_{x,y}$ on the right hand side (RHS) of Eq. (6) have (dimensionless) Fourier momentum and frequency components $\pm 1, \pm\omega$. The steady state solutions of $\theta_1, \phi_1$ are composed of these Fourier components only and therefore have the form

$$
\begin{aligned}
\theta_1(z,t) &= \theta_1^{(1,1)}e^{i(\omega t+z)} + \theta_1^{(1,-1)}e^{i(\omega t-z)} + h.c., \\
\phi_1(z,t) &= \phi_1^{(1,1)}e^{i(\omega t+z)} + \phi_1^{(1,-1)}e^{i(\omega t-z)} + h.c.,
\end{aligned}
\tag{8}
$$

which translate physically to two traveling waves running up or down the helix, depending on the relative sign in $\omega t \pm z$. The analytical forms of the pre-factors $\theta_1^{(1,\pm 1)}, \phi_1^{(1,\pm 1)}$ (without dipolar interactions) are given in App. B.

For circular polarized driving, $b_x = \pm b_y$, only one of the traveling wave modes gets excited. When dipolar interactions are switched on this only remains true if the demagnetization factors $N_x, N_y$ (see App. C for definition) in the plane perpendicular to $\mathbf{q}$ are identical. Right and left polarized circular driving are defined as the magnetic field rotating anticlockwise and clockwise in time, respectively, when we position ourselves at the origin and look in the positive $\hat{\mathbf{z}}$ direction. If we drive with a right polarized magnetic field $b_x = b_y$ only the down-traveling ($\omega t + z$) wave will be excited, and vice versa for left polarized driving.

To second order, $\mathcal{O}(\epsilon^2)$, the coupled equations in Eq. (7) have driving terms with Fourier components $k = 0, \pm 2, \omega' = 0, \pm 2\omega$. Here the mode $k = 0$, $\omega' = 0$ is special as it couples

to the Goldstone mode of the system, arising from the spontaneously broken translational symmetry of the conical state. Therefore, we can expect a *diverging* response. If we substitute the naïve choice of the $(k = 0, \omega' = 0)$-Fourier modes $\theta_2^{0,0}, \phi_2^{0,0}$, independent of $t, z$ into the first equation of (7), we quickly run into trouble, as the left and right sides of the equation do *not* balance each other. Mathematically, this conundrum can be solved by assuming that $\phi_2$ obtains a correction linear in $t$

$$\phi_2(z, t) = \phi_2^{\text{osc}}(z, t) + \omega_{\text{screw}} t. \tag{9}$$

Physically, this term does not describe an instability of the system but a rotation of the helix with angular velocity $\omega_{\text{screw}}$ which induces a screw-like motion. As shown in Fig. 1(b), individual spins precess rapidly with the driving frequency $\Omega$ (small circles in Fig. 1(b)). In analogy to the physics of a spinning top, this rapid local motion triggers a slow net precession of all spins around the **q** axis giving rise to a rotation of the helix with frequency $\omega_{\text{screw}}$. Equivalently, this screw-like rotation can also be interpreted as a translation of the helix in space parallel to **q** with constant velocity,

$$\mathbf{v}_{\text{screw}} = \hat{\mathbf{q}} \frac{\Omega_{\text{screw}}}{q}, \qquad \Omega_{\text{screw}} = \frac{D^2 |\gamma|}{J M_0} \omega_{\text{screw}}. \tag{10}$$

Within our perturbation theory $\omega_{\text{screw}}$ is quadratic in the oscillating fields. An analytic formula for $\omega_{\text{screw}}$ is given in App. B. In the absence of a constant magnetic field, $B_0 = 0$ and $\theta_0 = \pi/2$, we obtain

$$\omega_{\text{screw}} = \frac{\omega \left[ (b_R^2 - b_L^2)\left( (\alpha^2 + 1)\omega^2 + 4 \right) \right]}{8 \left[ (1 + \alpha^2)^2 \omega^4 + (5\alpha^2 - 4)\omega^2 + 4 \right]} \approx \frac{3\sqrt{2}}{32} \frac{b_R^2 - b_L^2}{(\omega - \sqrt{2})^2 + 9\alpha^2/4}, \tag{11}$$

where $b_{R/L} = b_x \pm b_y$ are the amplitudes of the right- and left polarized oscillating magnetic field. In the second line of Eq. (11) we expanded around the resonance frequency $\omega_{\text{res}} = \sqrt{2} + O(\alpha^2)$ in the limit of small damping $\alpha$.

Translating this back to physical units, Eq. (11) reads

$$\Omega_{\text{screw}} \approx \Omega_{\text{res}} \frac{3}{32} \frac{\gamma^2 (B_R^2 - B_L^2)}{(\Omega - \Omega_{\text{res}})^2 + 9\alpha^2 \Omega_{\text{res}}^2/8}, \tag{12}$$

where $\Omega_{\text{res}} = \sqrt{2} \frac{D^2 |\gamma|}{J M_0}$.

In Fig. 2(a) we show $\omega_{\text{screw}}$ as function of the (dimensionless) driving frequency $\omega$ for a vanishing external field. Switching from left- to right-polarized oscillating B-fields changes the sign of $\omega_{\text{screw}}$. At the resonance frequency of the the helix $\omega_{\text{screw}}$ is strongly enhanced in the limit of weak damping by the factor $1/\alpha^2$. For linear polarization $b_y = 0$ or, equivalently, $b_R = b_L$, there is no rotation of the helix, $\omega_{\text{screw}} = 0$, as predicted by Eq. (11). This changes when a static magnetic field parallel to **q** is switched on, see Fig. 2(b). In the resulting conical state the response to right- and left-polarized fields become different, see App. B, and one also obtains a finite result for linearly polarized fields oscillating only in the $x$ direction. In this case one can control the sign of $\omega_{\text{screw}}$ by changing the direction of the field **B**$_0$.

All formulas above are given in the absence of dipolar interactions. If they are included, an analytical calculation is still possible but the resulting formulas are too long to be displayed. The analytical result is plotted in Fig. 2(c) for vanishing external field (helical state) and in Fig. 2(d) for finite external field (conical state). While for vanishing external field the dipolar interactions mainly shift the resonance frequency, a qualitatively new effect occurs when both a static external field **B**$_0$ and dipolar interactions are considered together. In this case the resonance splits into a right-handed and a left-handed mode which selectively couple to the

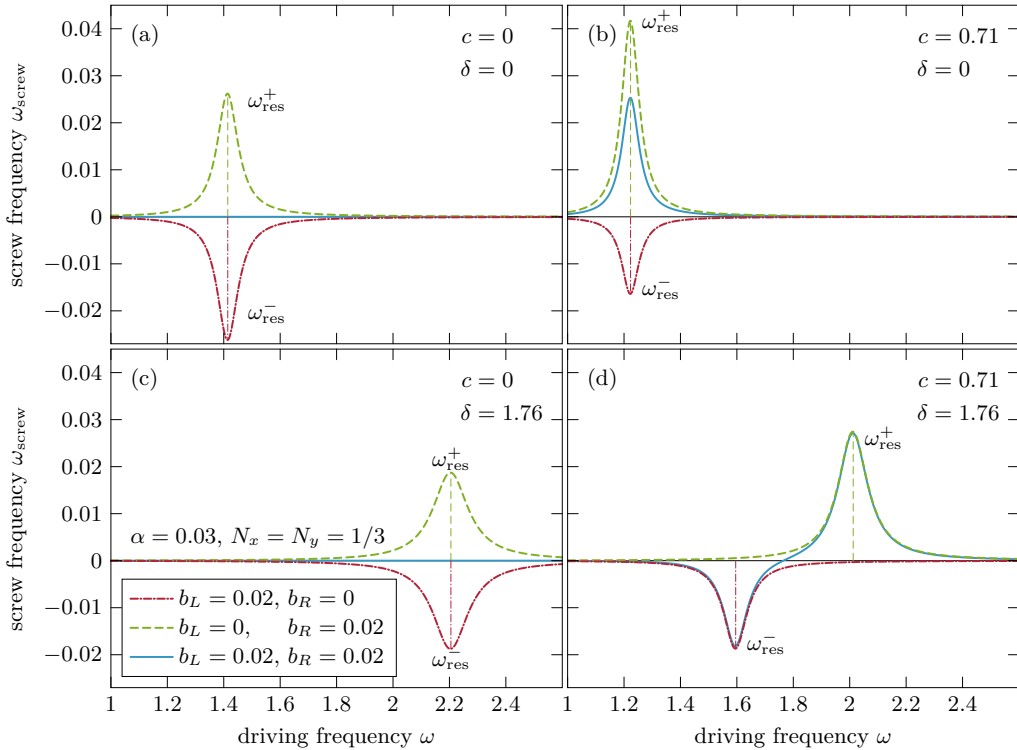

Figure 2: Dimensionless rotation frequency $\omega_{\text{screw}}$ of the magnetic texture plotted as a function of $\omega$ for different polarizations of driving magnetic field: left-circular polarized in red/dashed-dotted, right-circular polarized in green/dashed, and linearly polarized in blue/solid (with $\alpha = 0.03, N_x = N_y = 1/3, \gamma < 0$). Panels (a) and (b) discuss the case without dipolar interactions ($\delta = 0$), exactly described by Eq. (37), while dipolar interactions are included in panels (c) and (d), with $\delta = 1.76$, see App. C and Eq. (40) for the approximate behavior of $\omega_{\text{screw}}$ near resonance. In the absence of a static magnetic field, $\mathbf{B}_0 = 0$ ($c = 0$), panels (a) and (c), the left and right polarized contributions are equal and opposite and cancel each other when we drive with a linearly polarized driving field (blue curve). In finite field, panels (b) and (d), one can induce a rotation even for linearly polarized fields. In the presence of dipolar interactions and static field, the resonance splits, see App. C, and the sign of $\omega_{\text{screw}}$ can be controlled by changing frequencies.

right- and left- polarized oscillating fields. If in this situation a linearly polarized oscillating field is considered, $b_y = 0$, one can control the sign of $\omega_{\text{screw}}$ by changing the frequency of the applied field, see Fig. 2(d).

To confirm our results, we employ micromagnetic simulations. Using mumax3 [24,25], we solve the LLG Eq. (4) numerically for a conical state driven by an oscillating magnetic field. Parameters are chosen to describe $Cu_2OSeO_3$, where we choose the damping parameter to be $\alpha = 0.03$. For a quantitative comparison between simulations and analytical calculations (both including the effects of demagnetization fields), we determine $\Omega_{\text{screw}}$ as a function of driving frequency $\Omega$. From a set of simulations with different excitation frequencies $\Omega$, we extract $\Omega_{\text{screw}}$ as the linear slope of the azimuthal angle $\phi(t)$ of a single spin. For the chosen parameters rotation frequencies $\Omega_{\text{screw}}$ are in the MHz range, for driving frequencies in the GHz range. In Fig. 3 we compare the numerical result to the analytical formula and find an excellent agreement.

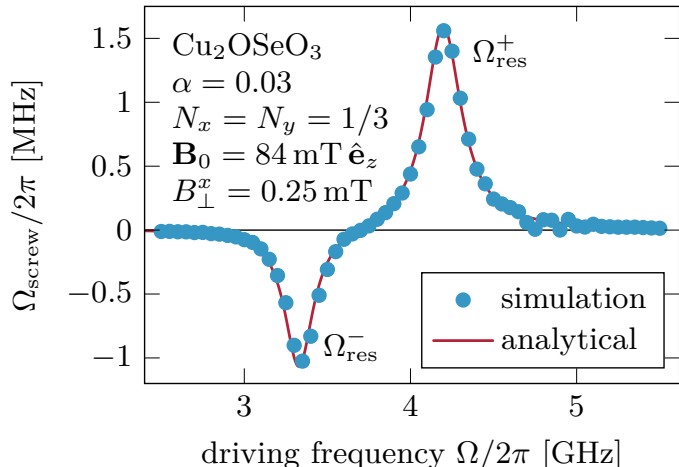

Figure 3: $\Omega_{\text{screw}}$ as a function of $\Omega$ from simulations and analytical calculations. The parameters $\gamma = -1.76 \times 10^{11}\,\text{T}^{-1}\,\text{s}^{-1}$, $J = 7.09 \times 10^{-13}\,\text{J}\,\text{m}^{-1}$, $D = 7.42 \times 10^{-5}\,\text{J}\,\text{m}^{-2}$, $M_0 = 1.04 \times 10^5\,\text{A}\,\text{m}^{-1}$ and $\alpha = 0.03$ have been chosen to describe $Cu_2OSeO_3$.

## 4 Floquet spin wave theory

As we will discuss below, the Archimedean screw solution becomes unstable when the driving fields get too large. This motivates us to investigate the stability of our solution using spin wave theory, or, more precisely, the "Floquet" variant of spin wave theory, which can be used to describe periodically driven systems. For this we have to expand the magnetization around the (perturbative) solution (5), $\mathbf{M} = \mathbf{M}_{\text{screw}} + \delta\mathbf{M}$ to derive an equation for $\delta\mathbf{M}$. In the following we use a notation similar (but not identical) to the one which is familiar from the Holstein-Primakoff treatment of quantum spins in the large $S$ limit [26]. Importantly, we will also include the effects of the phenomenological damping $\alpha$, which cannot easily be described by a quantum Hamiltonian. The magnetization is parametrized by

$$\mathbf{M} = M_0 \left( \mathbf{e}_3 (1 - a^* a) + \mathbf{e}_- a + \mathbf{e}_+ a^* \right), \tag{13}$$

where $a(\mathbf{r}, t)$ and $a^*(\mathbf{r}, t)$ are complex space- and time-dependent expansion coefficients. We use a coordinate system where $\mathbf{e}_3$ points parallel to the local magnetization of the Archimedean screw solution while $\mathbf{e}_{\mp}$ are perpendicular, with

$$\mathbf{e}_3 = \begin{pmatrix} \sin(\theta)\cos(\phi) \\ \sin(\theta)\sin(\phi) \\ \cos(\theta) \end{pmatrix}, \qquad \mathbf{e}_{\mp} = \frac{1}{\sqrt{2}} \begin{pmatrix} \cos(\theta)\cos(\phi) \pm i\sin(\phi) \\ \cos(\theta)\sin(\phi) \mp i\cos(\phi) \\ -\sin(\theta) \end{pmatrix}, \tag{14}$$

where the angles $\theta(\mathbf{r}, t)$ and $\phi(\mathbf{r}, t)$ are given by the solutions (5) discussed in Sec. 3. The expansion coefficients $a$ and $a^*$ have Poisson brackets $\{a(\mathbf{r}), a^*(\mathbf{r}')\} = \delta(\mathbf{r} - \mathbf{r}')$ which guarantees that $\{\hat{M}_i(\mathbf{r}), \hat{M}_j(\mathbf{r}')\} = i\epsilon_{ijk}\hat{M}_k(\mathbf{r})\delta(\mathbf{r} - \mathbf{r}')$ to leading order in a Taylor expansion in $a$. Using the notation of classical Hamiltonian dynamics, the LLG equation (4) takes the form

$$\text{sgn}(\gamma)\dot{\mathbf{M}} = i|\gamma|\{F, \hat{\mathbf{M}}\} - \alpha\hat{\mathbf{M}} \times \dot{\mathbf{M}}. \tag{15}$$

We will only be interested in terms linear in $a$ and $a^*$ and up to quadratic order in the oscillating external fields. More precisely, we consider to quadratic order only the contributions giving rise to the screw-like motion, omitting tiny oscillating terms at frequencies $2\omega$. A useful check of the expansion (and the Archimedean screw solution of Sec. 3) is that all constant terms

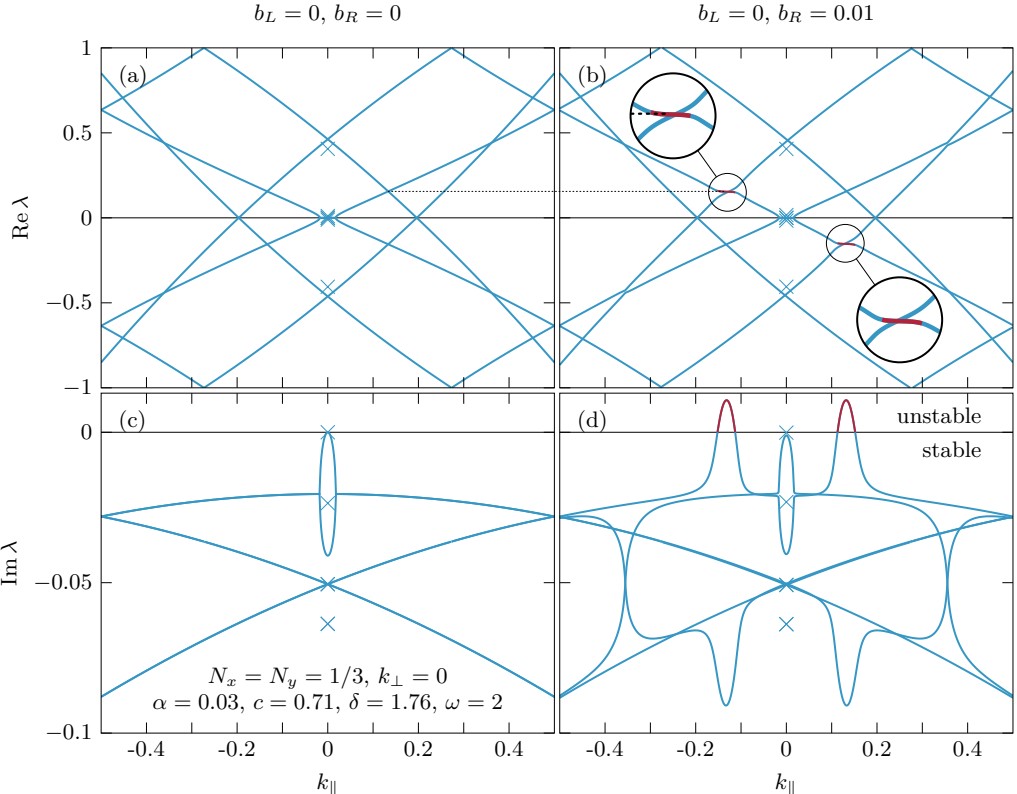

Figure 4: Eigenvalues $\lambda_{\mathbf{k}}$ of $M^F$ as a function of $k_{\parallel}$, the component of $\mathbf{k} \parallel \mathbf{q}$, for a system with parameters $\alpha = 0.03$, $c = 0.71$, $\delta = 1.76$, $N_x = N_y = N_z = 1/3$, $k_{\perp} = 0$. All graphs are in the first Brillouin zone $-q/2 < k_{\parallel} < q/2$. The real parts of the eigenfrequencies $\text{Re}[\lambda]$ are plotted in the first Floquet zone, between $-\omega/2 < \text{Re}[\lambda] < \omega/2$ with $\omega = 2$. The two graphs in the left column are the real and imaginary parts of $\lambda$ for an undriven system. All imaginary parts are negative, indicating that the conical static state is stable, as expected. The two graphs in the right column show the band energies for a driven system where the driving magnetic field is left circular polarized: $b_L = 0.01$, $b_R = 0$, with driving frequency $\omega = 2$ very close to the resonance frequency $\omega_{\text{res},+}$. The parts of the imaginary spectrum highlighted in red are unstable, and occur for $k_{\parallel} \sim 0.13q$, which corresponds to a $\text{Re}[\lambda] = \pm 0.16$. The crosses denote the spectrum at $k = 0$, which differs from the spectrum for $k \to 0$ due to the long-ranged dipolar interactions.

$O(a^0)$ drop out. Projecting the resulting equation onto the directions $\mathbf{e}_{\mp}$, see App. D, gives rise to

$$
\begin{aligned}
\dot{a} &= \frac{i(\text{sgn}(\gamma) - i\alpha)}{1 + \alpha^2}\{F^{(2)}, a\} - i\dot{\phi}\cos(\theta)a, \\
\dot{a}^* &= \frac{i(\text{sgn}(\gamma) + i\alpha)}{1 + \alpha^2}\{F^{(2)}, a^*\} + i\dot{\phi}\cos(\theta)a^*,
\end{aligned}
\tag{16}
$$

where $F^{(2)}$ is the contribution to $F$ quadratic in $a$ and $a^*$, and the factor of $|\gamma|$ has been absorbed into $F^{(2)}$.

The fact that we have periodic driving and are expanding around a state that is — in a frame of reference co-moving with our Archimedean screw — periodic in space and time makes Eq. (16) an ideal candidate for a Floquet treatment. We begin by defining the space

and time Fourier transformed fields $\tilde{a}_k^m, \tilde{a}_k^{m*}$ as

$$\tilde{a}_{\mathbf{k}}^m = \int dt \int d^3 r e^{im\omega t + i\mathbf{k}.(\mathbf{r}+\mathbf{v}_{\text{screw}}t)} a(\mathbf{r}),$$

$$\tilde{a}_{-\mathbf{k}}^{-m*} = \int dt \int d^3 r e^{im\omega t + i\mathbf{k}.(\mathbf{r}+\mathbf{v}_{\text{screw}}t)} a^*(\mathbf{r}). \tag{17}$$

Note the factor $\mathbf{r} + \mathbf{v}_{\text{screw}}t$ arising from our comoving coordinate system. Within our perturbative scheme, only the fields $\tilde{a}_{\mathbf{k}+n\mathbf{q}}^m, \tilde{a}_{\mathbf{k}+n\mathbf{q}}^{m*}$ with indices $m = -1, 0, 1$ and $n = -1, 0, 1$ couple to each other. We collect those in a 18-component vector $\Psi_{\mathbf{k}}^F$. The restriction of the Floquet space is formally justified because we investigate the system in the limit of small $B_\perp$ and our results for eigenenergies and decay rates are formally exact to quadratic order in $B_\perp$. Here it is important to realize that to conserve the Poisson brackets of the fields, one has to perform Bogoliubov transformations to diagonalize the dynamical matrix describing our system. Taking this into account, it is possible to recast Eq. (16) as a $18 \times 18$ matrix equation (see App. D for details)

$$\lambda_{\mathbf{k}} \Psi_{\mathbf{k}}^F = M_{\mathbf{k}}^F \Psi_{\mathbf{k}}^F, \tag{18}$$

with $\Psi_{\mathbf{k}}^F(t) = e^{-i\lambda t} \Psi_{\mathbf{k}}^F$. Importantly, the Floquet-Bogoliubov matrix $M^F$ is *not* a Hermitian matrix, both because of the underlying Bogoliubov transformation and the damping terms. Its eigenvalues are therefore complex in general,

$$\lambda_{\mathbf{k}} = \text{Re}[\lambda_{\mathbf{k}}] + i\text{Im}[\lambda_{\mathbf{k}}].$$

There is a clear physical interpretation for the real and imaginary parts of $\lambda$: the real part gives the temporal frequency of oscillation of the spin wave, whereas the imaginary part determines how fast it grows or decays in time. Importantly, the *sign* of the imaginary part determines whether the spin wave decays (negative imaginary part) or grows (positive imaginary part) exponentially in time, signaling an instability. As we show below, such instabilities are quite common in driven bosonic systems.

To understand how the oscillating fields affect the spin wave spectrum it is useful to consider first the case *without* oscillating fields, $B_\perp = 0$, shown in the left panels of Fig. 4. The upper left panel shows the real parts of the eigenmodes, $\text{Re}[\lambda_{\mathbf{k}}]$, as function of the momentum $k_\parallel$ parallel to the $\mathbf{q}$ direction. They always come in pairs $\pm\text{Re}[\lambda_{\mathbf{k}}]$ within the Bogoliubov formalism. Within the Floquet formalism all energies are 'folded back' to the first Floquet zone, $-\frac{\omega}{2} \le \text{Re}[\lambda_{\mathbf{k}}] < \frac{\omega}{2}$, i.e., they are calculated modulo the driving frequency $\omega$. The lower left panel displays the imaginary parts, $\text{Im}[\lambda_{\mathbf{k}}]$, which in the absence of driving are always negative and describe the decay of modes due to the damping term $\alpha$. Note that for $\mathbf{k} \to 0$ the Goldstone mode becomes overdamped and purely diffusive: the real part vanishes and $\lambda_{\mathbf{k}} \sim -i\alpha k_\parallel^2$. This is the behavior expected for Goldstone modes in systems where translational symmetry is spontaneously broken but where at the same time the underlying model lacks momentum conservation [27–29].

The panels on the right of Fig. 4 show how a finite oscillating field modifies the spin wave spectrum. Here the most dramatic effect occurs for the imaginary parts in the lower right panel: when the oscillating field is sufficiently large, they change sign and become positive. Thus the system becomes unstable when the oscillating field increases. The physics of the instability can be traced back to a resonance described by a simple $2 \times 2$ matrix

$$M_{\text{res}} \approx \begin{pmatrix} \epsilon_{i,\mathbf{k}}^0 - i\alpha\Gamma_i & \mu_\omega^{(1)} \\ -\mu_\omega^{(2)} & -\epsilon_{j,-\mathbf{k}}^0 + \omega - i\alpha\Gamma_j \end{pmatrix}. \tag{19}$$

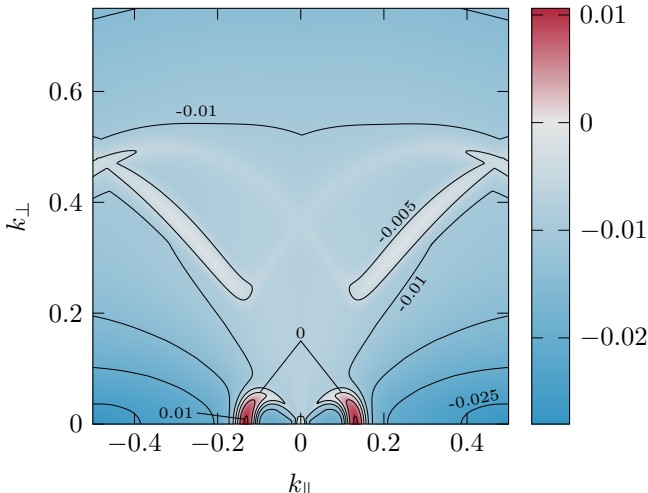

Figure 5: Largest $\text{Im}[\lambda_{\mathbf{k}}]$ plotted as a function of $\mathbf{k} = (k_\parallel, k_\perp)$ in the first Brillouin zone $1/2 \le k_\parallel/q < 1/2$, for $\omega = 2, b_x = 0.02, b_y = 0, \alpha = 0.03, c = 0.71, \delta = 1.76$. Regions where $\text{Im}[\lambda_{\mathbf{k}}] > 0$ are red, indicating an instability, regions where $\text{Im}[\lambda_{\mathbf{k}}] = 0$ are white, indicating a system on the verge of becoming unstable, and blue regions have $\text{Im}[\lambda_{\mathbf{k}}] < 0$, indicating that the Archimedean screw solution is stable there. The largest instability occurs along $k_\perp = 0$, at $k_\parallel/q = \pm 0.13$.

Here $\epsilon^0_{i\mathbf{k}} > 0$ denotes the energies of spin waves with band index $i$ of the unperturbed system and $\alpha\Gamma_i$ are the corresponding lifetimes. The frequency-dependent prefactors $\mu^{(i)}_\omega$ describe how the oscillating fields couple the energy level on the diagonal of the matrix. The coupling is most efficient when the driving frequency hits a $\mathbf{k} = 0$ resonance of the helix. Schematically, we find

$$\mu^{(1)}_\omega \mu^{(2)}_\omega \sim \frac{b_\perp^2}{(\omega - \omega_{\text{res}})^2 + (\alpha\Gamma)^2}. \tag{20}$$

The instability is most pronounced when

$$\epsilon^0_{i,\mathbf{k}} + \epsilon^0_{j,-\mathbf{k}} = \omega. \tag{21}$$

In this case the oscillating field can resonantly create a pair of spin waves out of the vacuum. In contrast, we do not find instabilities at energies $\epsilon^0_{i,\mathbf{k}} - \epsilon^0_{j,\mathbf{k}} = \pm\omega$ when spin waves are resonantly coupled. At this spin wave-creation resonance, the eigenvalues of $M_{\text{res}}$ are given by

$$\lambda^\pm_{\text{res}} = \epsilon^0_{i,\mathbf{k}} - i\alpha\frac{\Gamma_1 + \Gamma_2}{2} \pm i\sqrt{\mu^{(1)}_\omega \mu^{(2)}_\omega + \alpha^2 \left(\frac{\Gamma_1 - \Gamma_2}{2}\right)^2}. \tag{22}$$

Importantly, the sign of $\text{Im}[\lambda^+_{\text{res}}]$ changes when $b_\perp$ grows, signaling an instability. Assuming $\Gamma_1 \sim \Gamma_2 \sim \Gamma$ the system is only stable if (up to numerical prefactors)

$$b_\perp^2 \lesssim \left((\omega - \omega_{\text{res}})^2 + (\alpha\Gamma)^2\right)\alpha^2\Gamma^2. \tag{23}$$

More precisely, this formula is only valid for $\omega \approx \omega_{\text{res}}$. If one stays away from this point, then $\mu^{(i)}_\omega \sim b_\perp$ is independent of $\alpha$ and the system is only stable for

$$b_\perp \lesssim \alpha \, \text{const}. \tag{24}$$

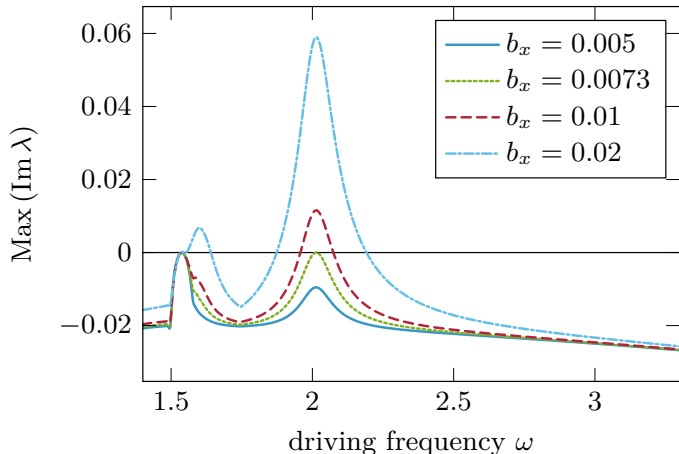

Figure 6: Largest Im[$\lambda_\mathbf{k}$] as a function of driving frequency $\omega$ obtained by diagonalizing the 18x18 matrix $M^F$ at the momentum of the leading instability, see Eq. (21) and Fig. 5, for increasing amplitudes of linearly polarized driving field $b_x$ ($c = 0.71, \delta = 1.76, \alpha = 0.03, N_x = N_y = 1/3$). For small oscillating fields, $b_x < 0.0073$ the system is stable for all frequencies, while it becomes unstable (Im[$\lambda_\mathbf{k}$] > 0) for larger fields, first close to the resonant frequencies. Further increasing the amplitude of the driving field increases the range of frequencies where instabilities occur.

In the limit $\alpha \to 0$ our calculation predicts that an arbitrarily weak oscillating field induces an instability. This is, however, an artifact of our approximation which ignores that the modes with finite energy and momentum can also decay via scattering processes. In this case an extra calculation of these lifetimes would be necessary to estimate when the instability occurs.

As a function of momentum, the resonance condition Eq. (21) is met along planes in momentum space. We therefore have to find the leading instability, i.e., the one where upon increasing $b_\perp$ the instability occurs first. In Fig. 5, Im[$\lambda_\mathbf{k}$] is shown as a function of $k_\perp$ and $k_\parallel$. At least for the parameter regime investigated by us, we find that the dominant instability occurs for $k_\perp = 0$.

To track the leading instability as function of frequency, we plot in Fig. 6 the largest Im[$\lambda$] as a function of driving frequency $\omega$, for a range of amplitudes of linearly polarized driving $b_x$. To produce this figure, we diagonalized $M^F$ at $k_\perp = 0$, choosing $k_\parallel$ to fulfil the resonance condition Eq. (21). As expected from Eq. (23) and (24), the system first becomes unstable at the resonance frequencies $\omega_{res}^\pm$ of the underlying conical state.

## 5 Formation of a time quasicrystal

The spin wave calculation rigorously shows that for $\alpha > 0$ the Archimedean screw solution is stable for small amplitudes of the oscillating field but becomes unstable upon increasing the field strength. However it cannot predict the fate of the unstable system. Therefore we again used numerical solutions of the LLG equation to analyze this regime. As the instability is expected to occur at finite momentum $k_\parallel$, it is essential to make the system sufficiently large in this direction. We therefore simulated a system with a length of up to 15 times the pitch of the helical state. In the perpendicular direction we use periodic boundary conditions using the fact that the instability occurs at $k_\perp = 0$, see Fig. 5. In very good agreement with our analytical solution, we find the stable Archimedean screw solution for small amplitudes

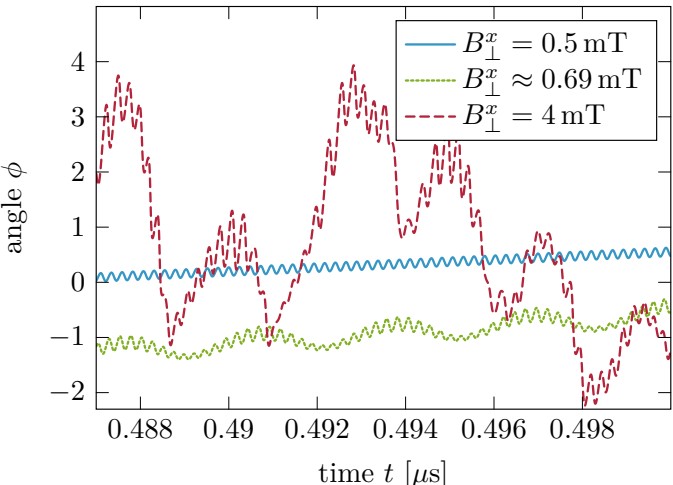

Figure 7: Angle $\phi(t)$ of a single spin as a function of time for increasing amplitude of the oscillating magnetic field, (parameters as in Fig. 3, with driving frequency $f = 4.15$ GHz), see also supplementary videos for an animated version. For small fields, $B_\perp^x = 0.5$ mT (solid blue line), the Archimedean screw solution is obtained. Fast oscillations of frequency $\omega$ trigger the screw-like motion of the conical state with frequency $\omega_{\text{screw}} \ll \omega$ giving rise to the finite average slope of $\phi(t)$. For larger field, $B_\perp^x = 0.69$ mT (dotted green line) a "time quasicrystal" forms, giving rise to a modulation with a frequency $f_{\text{new}} = 0.33$ GHz matching the instability predicted from spin wave theory ($\omega \approx 0.16$, red region in Fig. 4(b). For even stronger driving, $B_\perp^x = 4$ mT (dashed red line), one enters a chaotic regime, see also Fig. 8.

of the driving field as discussed above in Fig. 3. By increasing the amplitude of driving from $B_\perp^x = 0 - 1$mT in steps of 0.0625 mT, we obtain an instability around $B_\perp^x = 0.56 - 0.62$ mT ($b_x = 0.0075 - 0.0083$ in dimensionless units), see Fig. 8. This agrees well with the analytically predicted $b_{\text{crit}} = 0.0080$ for this set of parameters. Above this value we obtain — on top of the Archimedean screw solution — an extra modulation which has the spatial momentum $k = 0.13q$ and temporal angular frequency $\Omega_{\text{new}} = 2.1 \, \text{Grad} \, \text{s}^{-1}$ ($f_{\text{new}} = 0.33$ GHz) or $\omega_{\text{new}} = 0.16$ in our dimensionless units, see Fig. 7 and the supplementary videos. We thus find that momentum and frequency correspond exactly to the values where our Floquet analysis predicts the most unstable mode.

We can interpret this new mode as a kind of laser-type instability (or, equivalently, as a Bose-Einstein condensate) of the resonantly driven magnons. As the mode oscillates in time and space it defines a "time crystal", or more precisely, a "time quasicrystal" , as the frequency and momentum of oscillation determined by Eq. (21) are *incommensurate* with the driving frequency $\omega$ and the pitch vector $\mathbf{q}$ of the underlying conical state. From the viewpoint of symmetry, due to the presence of the oscillating field, time-translation invariance is only discrete. This discrete symmetry is then spontaneously broken by the time quasicrystal.

In Fig. 8 we show the screwing frequency as a function of the amplitude of the oscillating magnetic field. For small amplitudes, $\Omega_{\text{screw}}$ grows quadratically in $B_\perp^x$, following exactly the prediction of perturbation theory. The screwing frequency continues to grow in the regime where the time quasicrystal forms but the rate of growth is strongly reduced.

When we increase the driving further, the time quasicrystal also becomes unstable, see Fig. 8 and Fig. 7. We enter a chaotic regime discussed in more detail in App. E. Note that our simulations are not reliable in this regime as they assume translational invariance in the direction perpendicular to the $\mathbf{q}$ vector, which is valid both for the Archimedean screw solution

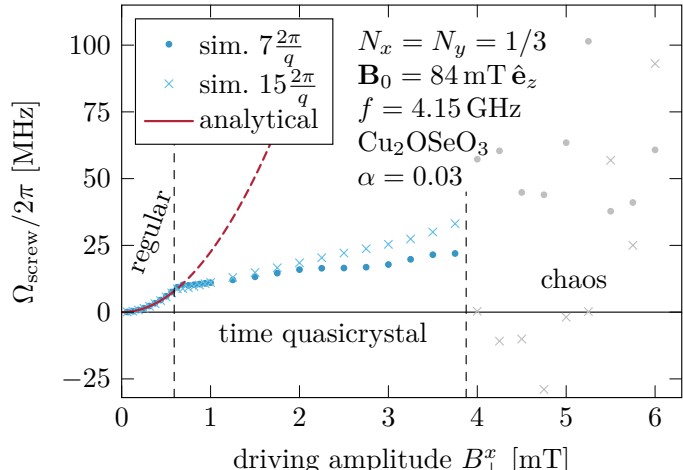

Figure 8: Screwing frequency as a function of the amplitude of the oscillating magnetic field, $B_\perp^x$ (parameters as in Fig. 3, with driving frequency $f = 4.15$ GHz) for two different sizes of the simulated system (7 and 15 times the pitch of the helix). For small fields $\Omega_{\text{screw}}$, the Archimedean screw solution is found, following the analytic prediction which high accuracy. Similarly, an instability resulting in the formation of a time quasicrystal occurs as predicted. The onset of the instability is slightly delayed for the smaller system, as the predicted wavelength of the instability $\lambda \approx 7.7 \frac{2\pi}{q}$ does not match the boundary conditions in this case. For $B_\perp^x \gtrsim 3.8$ mT we obtain chaotic solutions discussed in more detail in Appendix E).

and the time quasicrystal, but not in the chaotic regime.

## 6 Transport

Archimedean screws have been widely used for technological applications since antiquity, for example to transport water in irrigation systems, dehumidify low lying mines, or more recently even to deliver fish safely from one tank to another in so-called "pescalators" on fish farms. But could they also be used for transport in our system? In this section we want to show how coupling electrons to our rotating helical magnet gives rise to a finite DC current parallel to the **q** vector of the magnet. We model the electronic system by the following Hamiltonian

$$
\begin{aligned}
H &= H_s + H_{\text{dis}}, \\
H_s &= \int d^3r\, \mathbf{C}^\dagger(\mathbf{r})\left(\frac{\hat{\mathbf{p}}^2}{2m} + \lambda_{\text{so}}\hat{\mathbf{p}}\cdot\boldsymbol{\sigma} - J_H(\mathbf{n}(\mathbf{r},t)\cdot\boldsymbol{\sigma})\right)\mathbf{C}(\mathbf{r}), \\
H_{\text{dis}} &= \int d^3r\, V(\mathbf{r})\,\mathbf{C}^\dagger(\mathbf{r})\mathbf{C}(\mathbf{r}),
\end{aligned}
\tag{25}
$$

where $\mathbf{C}(\mathbf{r}) = \left(c_\uparrow(\mathbf{r}), c_\downarrow(\mathbf{r})\right)^T$ is a spinor containing the up and down components of the electron annihilation operators. In addition to the free energy term $\hat{\mathbf{p}}^2/2m$ we have a spin-orbit coupling term $\lambda_{\text{so}}\hat{\mathbf{p}}\cdot\boldsymbol{\sigma}$ and the exchange coupling $\mathbf{n}\cdot\boldsymbol{\sigma}$ of the electrons' spins to the local magnetization $\mathbf{n}$. For a static helix, the spin-orbit term induces the formation of exponentially flat mini-bands of periodicity $q$ in the $k_\parallel$ direction [30]. As we want to study the transport of electrons, it is essential to include the effects of disorder, which we model by a spin-independent random potential $V(\mathbf{r})$. In the following, we will model the effect of scattering from disorder by

a scattering rate $\frac{1}{\tau}$. We assume the following hierarchy of energy scales, $\epsilon_F > J_H \gg \frac{\hbar}{\tau}, \lambda_{so}\hbar k_F$, typical for magnets with weak spin-orbit coupling, where $\epsilon_F$ and $k_F$ are the Fermi energy and Fermi momentum, respectively.

We are interested now in a moving helix. We use a simplified Archimedean screw ansatz for $\mathbf{n}(\mathbf{r}, t)$

$$\mathbf{n} = \begin{pmatrix} \sin(\theta_0)\cos(qz - \omega_{screw}t) \\ \sin(\theta_0)\sin(qz - \omega_{screw}t) \\ \cos(\theta_0) \end{pmatrix}, \tag{26}$$

where we have suppressed all the oscillations which are multiples of the driving frequency $\omega$ and kept only the $\omega_{screw}$ time dependence.

In the absence of disorder (and also in the absence of Umklapp scattering due to electron-electron interactions), the problem can be solved by moving to a frame of reference comoving with the helix using the transformation $\mathbf{C}^\dagger(\mathbf{r}) \to \mathbf{C}^\dagger(\mathbf{r} - \mathbf{v}_{screw}t)$. The current in the comoving frame vanishes and therefore the electronic current density $j_\parallel$ in the lab frame is simply given by

$$\langle j_\parallel \rangle = e v_{screw}(n_\uparrow + n_\downarrow), \tag{27}$$

where $n_{\uparrow/\downarrow}$ are the electron densities of majority and minority electrons, respectively.

More realistically, one has to take into account the effects of disorder (or Umklapp scattering) which is expected to dominate transport properties. Here it is useful to consider a transformation where (i) impurities do *not* move, and (ii) the Hamiltonian is diagonal in the dominant term $J_H$, i.e. the spins of the electrons are aligned and anti-aligned with the time-dependent local magnetization $\mathbf{n}(\mathbf{r}, t)$. This can be achieved by rotating the spin-quantization axis using the unitary matrix $U$

$$U(\mathbf{r}, t) = \begin{pmatrix} \cos(\theta_0/2) & \sin(\theta_0/2)e^{-i\phi} \\ \sin(\theta_0/2)e^{i\phi} & -\cos(\theta_0/2) \end{pmatrix}, \tag{28}$$

where $\phi = qz - \omega_{screw}t$. We can then define $\mathbf{C}(\mathbf{r}) = U(\mathbf{r})\mathbf{D}(\mathbf{r})$, such that $d_\uparrow^\dagger, d_\downarrow^\dagger$ now create electrons with spins parallel and anti-parallel to the local time-dependent magnetization $\mathbf{n}$, respectively. Rewriting Eq. (25) in terms of $\mathbf{D}, \mathbf{D}^\dagger$ and switching to Fourier space we obtain approximately

$$\tilde{H} \approx \sum_{\sigma,\mathbf{k}} \epsilon_{\sigma,\mathbf{k}} d_{\sigma,\mathbf{k}}^\dagger d_{\sigma,\mathbf{k}} + H_1(t) + H_{dis}, \tag{29}$$

$$H_1(t) = \sum_{\sigma,\mathbf{k}} \frac{\hbar s k_\perp \lambda_{so}}{2} (d_{\sigma,\mathbf{k}}^\dagger d_{\sigma,\mathbf{k}+\mathbf{q}} e^{-i\omega_{screw}t} + h.c.), \tag{30}$$

$$\epsilon_{\uparrow/\downarrow,\mathbf{k}} \approx \frac{\hbar^2}{2m}\left((k_\parallel \mp k_0)^2 + k_\perp^2\right) \mp J_H,$$

$$k_0 = \frac{(1-c)q}{2} - \frac{cm\lambda_{so}}{\hbar}, \quad s = \sin(\theta_0), \quad c = \cos(\theta_0).$$

Here we ignored some small static correction terms to $\epsilon_{\sigma,\mathbf{k}}$ as well as spin-mixing terms of type $d_\uparrow^\dagger d_\downarrow$, which can be ignored because of the large splitting between the minority and majority-spin Fermi surfaces due to $J_H$. Importantly, the unitary transformation does not affect the potential scattering term $H_{dis}$.

Following the rotation by $U$, the only time-dependent term $H_1(t)$ in the Hamiltonian comes from spin orbit interactions. For $H_1 = 0$, we obtain electrons with $\epsilon_{\sigma\mathbf{k}}$ describing majority and minority electrons. Their Fermi surfaces are shifted by $k_\parallel = \pm k_0$ both due to the spin-orbit interactions and the rotation of the spins by the matrix $U$.

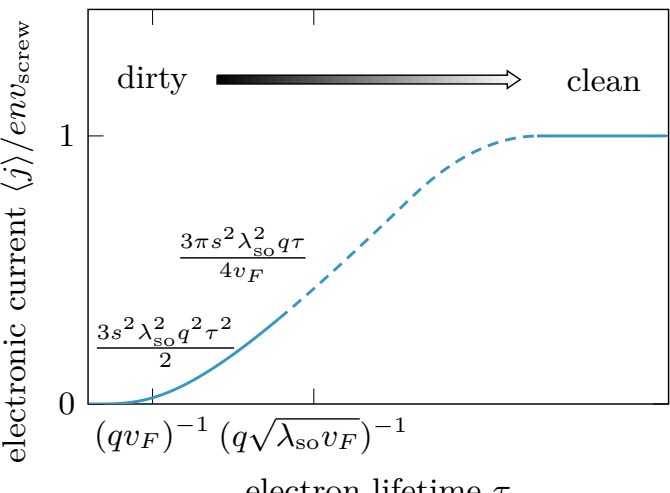

Figure 9: Schematic plot of the electronic current density $\langle j_\parallel \rangle$ as a function of electron lifetime $\tau$. Note that we have suppressed the spin indices, which is justified for a strongly spin polarized system $N_\uparrow \gg N_\downarrow$. For a strongly disordered system, when $\tau \ll (q v_F)^{-1}$, $\langle j_\parallel \rangle = \frac{3s^2\lambda_{\mathrm{so}}^2 q^2 \tau^2}{2}$ is quadratic in $\tau$. In the range $(q v_F)^{-1} \ll \tau \ll (q\sqrt{v_F})^{-1}$, $\langle j_\parallel \rangle$ grows linearly with $\tau$. Our perturbative assumptions break down in the dashed region, but we know that for a very clean system with no disorder ($\tau \gg 1$) the current must plateau at $\langle j_\parallel \rangle = e n v_{\mathrm{screw}}$.

We would like to evaluate the expectation value of the parallel component of the current operator

$$J_\parallel = -\frac{e\hbar}{m} \sum_{\mathbf{k}} (k_\parallel - k_0) d^\dagger_{\uparrow,\mathbf{k}} d_{\uparrow,\mathbf{k}} + (k_\parallel + k_0) d^\dagger_{\downarrow,\mathbf{k}} d_{\downarrow,\mathbf{k}}, \tag{31}$$

treating $H_1(t)$ as a small time dependent perturbation. We can formulate this as a Keldysh problem

$$\langle J_\parallel(t) \rangle = \left\langle U(-\infty, +\infty) T \left( U(+\infty, -\infty) \tilde{J}_\parallel(t) \right) \right\rangle, \tag{32}$$

where $U(t_2, t_1) = e^{-i\int_{t_1}^{t_2} \tilde{H}_1(t') dt'}$ and we denote as $\tilde{O}(t) = e^{iH_0 t} O e^{-iH_0 t}$ the operators in the interaction picture. Ultimately we are interested in the DC component of $J_\parallel$, which to lowest order comes in at second order in the perturbation $H_1(t) \sim e^{i\omega_{\mathrm{screw}} t}$. After some algebra (see App. F for technical details) we arrive at

$$\langle J_\parallel \rangle = \frac{2\lambda_{\mathrm{so}}^2 s^2 e\hbar^4 q v_{\mathrm{screw}}}{m} \sum_{\sigma,\mathbf{k}} \frac{k_\perp^2 (k_\parallel - \sigma k_0)(n_{\sigma,\mathbf{k}} - n_{\sigma,\mathbf{k+q}})(\epsilon_{\sigma,\mathbf{k}} - \epsilon_{\sigma,\mathbf{k+q}})}{\left( \left( \epsilon_{\sigma,\mathbf{k+q}} - \epsilon_{\sigma,\mathbf{k}} \right)^2 + (\hbar\tau^{-1})^2 \right)^2}, \tag{33}$$

where we have used that $\omega_{\mathrm{screw}} = q v_{\mathrm{screw}}$. Here, $n_{\sigma,\mathbf{k}}$ is the Fermi distribution function $(1 + e^{\beta(\epsilon_{\sigma,\mathbf{k}} - \epsilon_{\sigma,k_F})})^{-1}$. Performing the integral in $k$- space at $T = 0$ amounts to integrating over the two Fermi spheres located at $\pm k_0$ discussed earlier (see App. F for details). We obtain

$$\langle j_\parallel \rangle \approx \sum_{\sigma = \uparrow, \downarrow} e n_\sigma v_{\mathrm{screw}} \begin{cases} \frac{3s^2 \lambda_{\mathrm{so}}^2 q^2 \tau^2}{2}, & v_{F,\sigma} \tau \ll \frac{2\pi}{q} \\ \frac{3\pi s^2 \lambda_{\mathrm{so}}^2 q\tau}{4 v_{F,\sigma}}, & \frac{2\pi}{q} \ll v_{F,\sigma} \tau \ll \frac{\sqrt{v_{F,\sigma}}}{q\sqrt{\lambda_{\mathrm{so}}}} \end{cases}. \tag{34}$$

In the limit when the mean free path of the electrons $v_F \tau$ is smaller than the wavelength of the helix $\frac{2\pi}{q}$, the current is quadratically dependent on the electron's lifetime $\tau$. In the

opposite limit, $v_F\tau \gg 2\pi/q$, in contrast, the current is linear in $\tau$ and thus proportional to the conductivity of the system. Eq. (34) has been derived in perturbation theory in $\lambda_{so}$ and thus cannot describe the formation of band-gaps and minibands triggered by $\lambda_{so}$. These minibands have a band splitting of the order of $\Delta \sim \hbar q \sqrt{v_F \lambda_{so}}$ [30] and thus perturbation theory is only reliable for $\tau\Delta/\hbar \ll 1$ which sets an upper limit for the regime of validity of the second line in Eq. (34). These results are summarized in Fig. 9.

## 7  Experimental signatures and conclusions

Within our numerical and analytical calculations, we found that even for a weak oscillating magnetic field, the magnetic helix starts to rotate in a screw-like motion. This means that naïve perturbation theory breaks down as the difference, $\mathbf{M}(\mathbf{r},t) - \mathbf{M}_0(\mathbf{r})$, of the magnetization of the perturbed system, $\mathbf{M}(\mathbf{r},t)$, and of the unperturbed state, $\mathbf{M}_0(\mathbf{r})$, grows linearly in time. Physically this arises because the system couples to a Goldstone mode and technically it can be described by using the moving helix as a starting point of perturbation theory. Similar effects also arise in many other systems. For example, one can move skyrmions by oscillating fields [8, 10] and ratchets also work by a similar mechanism, see [31] for a theoretical review and [32, 33] for experiments.

Friction plays a decisive role for this phenomenon. Both the force which induces the rotation of the helix and the counter force arising from the motion of the helix are proportional to the friction coefficient. As a result, the frequency $\Omega_{screw}$ describing the screw-like rotation obtains a finite value in the limit of vanishing friction constant, $\alpha \to 0$. A second important effect is that friction is needed to stabilize the state and to avoid instabilities and the onset of chaos in this driven nonlinear system. The net effect is that one can reach larger values of $\Omega_{screw}$ in systems with stronger friction. Here both extrinsic friction (parametrized by $\alpha$) arising from coupling to phonons or electrons and intrinsic friction arising from magnon-magnon scattering play a role but only the first effect was included in our Floquet spin wave theory of the instabilities.

In experimental systems the role of pinning by disorder has to be considered. The following order-of-magnitude estimates are motivated by the parameters in MnSi, arguably the best investigated chiral magnet. In the presence of pinning, we expect that a critical strength of the oscillating field is required before the helix starts to move. To obtain a rough estimate, we assume a screw frequency $\Omega_{screw} \sim 10\,\text{MHz}$ (obtained using Eq. (10), for micromagnetic parameters $J = 7.05 \times 10^{-13}\,\text{J}\,\text{m}^{-1}$, $D = 2.46 \times 10^{-4}\,\text{J}\,\text{m}^{-2}$, $M_0 = 1.52 \times 10^5\,\text{A}\,\text{m}^{-1}$ corresponding to MnSi [15, 34, 35], as well as $\omega_{screw} = 0.0007$ for oscillating fields of the order of $0.5\,\text{mT}$ and $\alpha \sim 0.01$). For a helix with a pitch of $200\,\text{Å}$, this corresponds to a speed of $v_{screw} \approx 200\,\text{mm}\,\text{s}^{-1}$. We can compare this speed to the velocity of skyrmions driven by a current $j$, in MnSi [6]. Skyrmions are expected to have a very similar friction and pinning compared to the helical and conical states as the magnetization is modulated on the same length scale. They start to move above a critical current density, $j_c$, and their speed can be estimated from measurements of the Hall effect [6]. For example, at a current density of $2j_c$, the skyrmion velocity has been estimated to be about $0.2\,\text{mm/s}$, which is three orders of magnitude *smaller* than our estimate for $v_{screw}$. We conclude that at least for resonant driving one can likely induce the screw-like motion of the helix in materials with low pinning as realized in MnSi and similar materials.

To detect the rotation of the helix, one can try to pick up a signal from the rotating magnetization using, e.g., a detector on the surface of the crystal. A more intriguing approach would be to observe the Archimedean screw "in action". For example, in a metallic system we have shown that it can transport charge. We therefore expect that a voltage will build up parallel to

the orientation of the helix. The current and voltage will depend sensitively on the amount of disorder and the strength of spin-orbit coupling in the system. For example, in the chiral magnet CoGe spin-orbit interaction lead to a band-splitting of almost 10% of the bandwidth [36] and similar values are expected for MnSi. Thus we estimate $\lambda_{so}/v_F \sim 10^{-2} - 10^{-1}$. Furthermore, MnSi can be grown with exceptional crystal quality and residual resistivities well below $1\,\mu\Omega\,\mathrm{cm}$, giving rise to a mean free path larger than $1000\,\text{\AA}$ at low $T$ [37]. Assuming a mean free path of the order of the pitch of the helix and using $n \sim 4 \cdot 10^{22}\,\mathrm{cm}^{-3}$ [38] our calculation yields current densities of order $10^4 - 10^7\,\mathrm{A\,m}^{-2}$. Even for the smallest values in this range, the corresponding voltage building up in such a system will be very easy to detect. In good metals, however, the skin depth (the length scale on which electromagnetic fields penetrate the sample) is only of the order of $1\,\mu\mathrm{m}$ at microwave frequencies. Therefore one should either use thin samples or bad metals. An interesting alternative is to try to detect thermal gradients or gradients in the magnetization arising from the transport of heat and spin, respectively.

For stronger driving, we predict the formation of a 'time quasicrystal' arising in the driven system. This can probably be detected most easily by picking up the radiation arising from the oscillating magnetization which is expected to be in the $100\,\mathrm{MHz} - 1\,\mathrm{GHz}$ range. The detection of any monochromatic emission with a frequency smaller than the driving frequency is a unique signature of such a state.

In conclusion, we have shown that using the helical and conical states of chiral magnets and weak oscillating fields one can realize an Archimedean screw on the nanoscale. As one of the archetypal machines known to mankind, it can be used to explore the transport of charge, spin or heat in a novel setup.

# Acknowledgements

We thank Joachim Hemberger, Christian Pfleiderer, Andreas Bauer, Markus Garst and, especially, Yuriy Mokrousov for useful discussions. NdS also thanks S. Mathey and V. Lohani for helpful discussions.

**Author contributions**    NdS performed the analytical calculations, with support from AR. LH collected the numerical data. LH and AR designed the study. All three authors contributed to writing the article.

**Funding information**    We acknowledge the financial support of the DFG via SPP 2137 (project number 403505545) and CRC 1238 (project number 277146847, subproject C04). We furthermore thank the Regional Computing Center of the University of Cologne (RRZK) for providing computing time on the DFG-funded (Funding number: INST 216/512/1FUGG) High Performance Computing (HPC) system CHEOPS as well as support.

# A   Supplementary videos

The supplementary video files can be found at https://arxiv.org/src/2012.11548v4/anc.

The supplementary video `screw_schematic.mp4` gives an animated version of Fig. 1(b), showing the motion of spins in the regime where the Archimedean screw solution is realized. The second video, `time_crystal_schematic.mp4` shows a similar plot in the time quasicrystal phase. Finally, in the supplementary video `simulation_comparison.mp4` the dynamics of three different phases realized for $B_\perp = 0.5\,\mathrm{mT}, 1\,\mathrm{mT}$ and $4\,\mathrm{mT}$ (other parameters

are as in Fig. 8 of the main text) is shown. While the first two videos are schematic, the last video is based on simulation data. The color encodes changes in the tilt angle $\theta$ which is also shown in the blue curves. $\theta$ is regularly sinusoidal in the Archimedean screw phase, while in the time crystal phase it acquires an additional space and time component, which one can observe by following the slowly down-moving flat region in time. In the chaotic regime, a large number of modes are excited. Consequently, $\theta$ does not show such a clear pattern (see also App. E and Fig. 10).

## B Analytical formulas for the Archimedean screw

In this section we collect analytical results describing the Archimedean screw solution. Details on the derivations of the formulas in the presence of dipolar interactions can be found in Appendix C below.

Without dipolar interactions, the first order complex prefactors which solve Eq. (6) using ansatz (8) are

$$
\begin{aligned}
\theta_1^{(1,-1)} &= \frac{(b_x + b_y)(\omega(\text{sgn}(\gamma) - i\alpha c) - c)}{4((1+\alpha^2)\omega^2 + i\alpha(c^2-3)\omega + c^2 - 2)}, \\
\theta_1^{(1,1)} &= \frac{(b_y - b_x)(\omega(\text{sgn}(\gamma) + i\alpha c) + c)}{4((1+\alpha^2)\omega^2 + i\alpha(c^2-3)\omega + c^2 - 2)},
\end{aligned}
\tag{35}
$$

$$
\begin{aligned}
\phi_1^{(1,-1)} &= \frac{-i(b_x + b_y)\left(c^2 - 2 + \omega(\text{sgn}(\gamma)c - i\alpha)\right)}{4s((1+\alpha^2)\omega^2 + i\alpha(c^2-3)\omega + c^2 - 2)}, \\
\phi_1^{(1,1)} &= \frac{i(b_x - b_y)\left(c^2 - 2 - \omega(\text{sgn}(\gamma)c + i\alpha)\right)}{4s((1+\alpha^2)\omega^2 + i\alpha(c^2-3)\omega + c^2 - 2)},
\end{aligned}
\tag{36}
$$

with $s = \sin\theta_0$ and $c = \cos\theta_0$. Note that in the special cases $b_x = \pm b_y$, one of each pair of pre-factors $\theta_1^{(1,\pm1)}$ and $\phi_1^{(1,\pm1)}$ vanishes. Physically, $b_x = \pm b_y$ correspond to right and left circular polarized driving, respectively. Circular polarized driving couples only to one of the two modes. This motivates us to define

$$
\begin{aligned}
b_L &= b_x - b_y, \\
b_R &= b_x + b_y.
\end{aligned}
$$

In these new variables, we get left circular driving by setting $b_R = 0$, right circular driving by setting $b_L = 0$, and linearly polarized driving in the $x, y$ directions by choosing $b_L = \pm b_R$. Any other choice of $b_L, b_R$ corresponds to the general elliptical drive.

Without dipolar interactions, the two modes are degenerate with resonant frequency $\omega_{\text{res}} = \sqrt{2-c^2}$. To evaluate the screwing frequency we substitute Eq. (35) into the first equation of Eq. (7). Here $\partial_t \phi_2 = \omega_{\text{screw}}$ balances all the other DC components in the equation, which can be computed from the first order solutions ($\partial_t \theta_2$ and $\phi_2''$ do not contribute as they are both oscillating in time and/or space). We obtain

$$
\omega_{\text{screw}} = \frac{\omega\left[(b_R^2 - b_L^2)\left((\alpha^2+1)\omega^2 - 3c^2 + 4\right) - \text{sgn}(\gamma)2(b_R^2 + b_L^2)c\omega\right]}{8\left[(1+\alpha^2)^2\omega^4 + (2c^2 - 4 + \alpha^2(c^4 - 4c^2 + 5))\omega^2 + (c^2-2)^2\right]}.
\tag{37}
$$

When dipolar interactions are switched on (see App. C for details and definitions), the single resonance frequency $\omega_{\text{res}}$ gets shifted and split into two different resonance frequencies

$\omega_{\text{res}}^{\pm}$ proportionally to $\delta$, a dimensionless measure of the strength of the dipolar interactions

$$
\begin{aligned}
\omega_{\text{res}}^{\pm} = \frac{1}{2} \Bigg\{ \Bigg[ & c^2 \left( \delta^2 (2N_x N_y - N_x - N_y) - 4 - 4\delta \right) + (\delta + 2)(\delta(N_x + N_y) + 4) \\
& \pm \sqrt{\Big( \left( c^2 \left( \delta^2 (2N_x N_y - N_x - N_y) - 4 - 4\delta \right) + (\delta + 2)(\delta(N_x + N_y) + 4) \right)^2 } \\
& \overline{\phantom{\pm \sqrt{}} - 4 \left( c^2 \left( 2\delta + \delta^2 N_x + 2 \right) - (\delta + 2)(\delta N_x + 2) \right) \left( c^2 \left( 2\delta + \delta^2 N_y + 2 \right) - (\delta + 2)(\delta N_y + 2) \right) \Big)} \Bigg],
\end{aligned}
\tag{38}
$$

with

$$
\delta = \frac{\mu_0 J M_0^2}{D^2}.
\tag{39}
$$

The prefactors $\theta_1^{(1,1)}, \phi_1^{(1,1)}$ describing the linear-response solution with dipolar interactions are lengthy and therefore not listed here. If the shape of the crystal is cylindrically symmetric around the axis of the helix, $N_x = N_y$, circular polarized light couples only to a single mode. One can analytically calculate the screwing frequency to second order in the oscillating fields. Instead of showing the exact result of this lengthy calculation, which is too long, we display below the most singular contribution obtained from a Taylor expansion around the two resonance frequencies $\omega_{\text{res}}^{\pm}$

$$
\omega_{\text{screw}} \approx \mp \frac{\text{sgn}(\gamma) b_{\text{R/L}}^2 A_{\text{sgn}(\gamma)\pm}}{\left( \omega - \omega_{\text{res}}^{\text{sgn}(\gamma)\pm} \right)^2 + \Delta\omega^2}, \quad \Delta\omega^2 = \alpha^2 \frac{\left( \omega_{\text{res}}^{\text{sgn}(\gamma)\pm} \right)^2 \left( c^2 (5\delta + 6) - 7\delta - 18 \right)^2}{4(\delta + 6)\left( c^2 (5\delta + 6) - 6(\delta + 2) \right)}.
\tag{40}
$$

The prefactor $A_{\pm}$ turns out to be finite in the limit $\alpha \to 0$ and therefore $\omega_{\text{screw}} \propto 1/\alpha^2$ at resonance. If the system is driven with $|\omega - \omega_{\text{res}}^{\pm}| \gg \Delta\omega$, in contrast, $\omega_{\text{screw}} \sim (\omega - \omega_{\text{res}}^{\pm})^{-2}$ remains finite in the limit $\alpha \to 0$.

## C  Calculation of dipolar interactions

In contrast to the Heisenberg and DMI energy terms which are local, dipolar interactions are long-ranged. They are not only much more computationally costly to calculate but require different treatment for the case $k = 0$ (the so-called demagnetization field limit) and the limit $k \to 0$ in the thermodynamic limit [39]. Here the calculation for $k = 0$ has to take into account the energy stored in the magnetic fields outside of the sample.

### C.1  Demagnetization fields

Let us denote the $k = 0$ or DC component of the magnetization as $\overline{M}_i$ with

$$
\overline{M}_i = \frac{1}{V} \int d^3 r M_i(\mathbf{r}),
\tag{41}
$$

where $V$ is the total volume of the sample. For our helical or conical texture, this integral only needs to be done over the $z$-axis due to the translational invariance in the $x, y$ directions. Applying Eq. (41) to the static helical or conical ansatz Eq. (3), we find that $\overline{M}_x = \overline{M}_y = 0$ and the only non-zero DC component is $\overline{M}_z = M_0 \cos(\theta_0)$. In general, the magnetization of the system will set up internal demagnetization fields in directions opposite the applied

external magnetic fields. For a sample with an ellipsoidal shape, the mathematical expression for these internal demagnetization fields is particularly simple and given by

$$\mathbf{B}_{\text{demag}} = -\mu_0 \underline{\underline{N}} \cdot \overline{\mathbf{M}} = - \begin{pmatrix} N_x & 0 & 0 \\ 0 & N_y & 0 \\ 0 & 0 & N_z \end{pmatrix} \cdot \begin{pmatrix} \overline{M}_x \\ \overline{M}_y \\ \overline{M}_z \end{pmatrix}, \tag{42}$$

where $N_i$ are the demagnetization factors which solely dependent on the shape of the sample and obey the identity $\text{Tr}(\underline{\underline{N}}) = 1$. The corresponding contribution to the free energy Eq. (1) is

$$F_{\text{dip},k=0} = \frac{1}{2}\mu_0 (\overline{\mathbf{M}} \cdot \underline{\underline{N}} \cdot \overline{\mathbf{M}}) V. \tag{43}$$

Applying this to the static conical helix, we obtain an additional contribution $F_{\text{dip,k}=0} = \frac{1}{2}\mu_0 N_z$ $M_0^2 \cos^2(\theta_0)$. For the static conical state, $q$ is unaffected but $\cos(\theta_0) = \frac{b_0}{1+\delta N_z}$ changes. As $\delta, N_z > 0$, this means that $\theta_0$ increases. This is a consequence of the internal demagnetization field $\mathbf{B}_{\text{demag}}$ opposing and reducing the applied field $\mathbf{B}_0$.

For the dynamical calculation, we need to add Eq. (42) to $\mathbf{B}_{\text{eff}}$ in the RHS of Eq. (4), but now we need to substitute the dynamic ansatz Eq. (5) with Eq. (8) into $\overline{\mathbf{M}}$. This gives many new terms on the RHS of the first and second order equations Eq. (6) and (7). Here are the first order in $\epsilon$ contributions

$$\begin{aligned}
\overline{M}_x^{(1)} &= \frac{e^{i\omega t}}{2}\left[c(\theta_1^{(1,1)} + \theta_1^{(1,-1)}) - is(\phi_1^{(1,1)} - \phi_1^{(1,-1)})\right] + h.c., \\
\overline{M}_y^{(1)} &= \frac{e^{i\omega t}}{2}\left[ic(\theta_1^{(1,1)} - \theta_1^{(1,-1)}) + s(\phi_1^{(1,1)} + \phi_1^{(1,-1)})\right] + h.c., \\
\overline{M}_z^{(1)} &= 0.
\end{aligned} \tag{44}$$

Using this result, one can calculate the corresponding magnetic fields using Eq. (42) which contribute to the effective magnetic field in the LLG equation, Eq. (4). Mathematically, the oscillating finite-$k$ contributions of $\theta_1(z,t), \phi_1(z,t)$ modify the $k=0$ magnetization because we are Taylor expanding around a spatially modulated static helix. At second order $\epsilon^2$ we have many more terms because there are more combinations between the perturbing terms and static solution which modify the $k=0$ magnetization. We do not list them here as they are lengthy, but the procedure to obtain them follows exactly from that used for the first order terms Eq. (44).

## C.2 Finite $k$ contributions

At finite momentum, for $k$ larger than the inverse system size, the contributions of the dipolar interactions to the free energy take the form

$$F_{\text{dip},k\neq0} = \frac{1}{2}\mu_0 V \sum_{\mathbf{k}\neq0} \frac{(\mathbf{M_k} \cdot \mathbf{k})(\mathbf{M_{-k}} \cdot \mathbf{k})}{k^2}, \tag{45}$$

where $\mathbf{M}_k = \frac{1}{V}\int d^3 r \mathbf{M}(r) e^{-i\mathbf{k}\cdot\mathbf{r}}$. We will first analyze how this term affects the static conical helix. The Fourier transform of Eq. (3) is

$$\mathbf{M_k} = M_0 \begin{pmatrix} \frac{1}{2}\sin(\theta_0)[\delta(\mathbf{k}-\mathbf{q}) + \delta(\mathbf{k}+\mathbf{q})] \\ \frac{1}{2}\sin(\theta_0)[\delta(\mathbf{k}-\mathbf{q}) - \delta(\mathbf{k}+\mathbf{q})] \\ \cos(\theta_0)\delta(\mathbf{k}) \end{pmatrix}, \tag{46}$$

therefore, since the only non-zero Fourier components of magnetization $\mathbf{M}_{\pm q} \perp \mathbf{q}$, $F_{\mathrm{dip},k\neq 0} = 0$ for the static conical helix. Thus only the DC $k = 0$ components of magnetization play a role in the determination of $q, \theta_0$ for the static conical helix.

Moving on to dynamics, we need to extract a magnetic field

$$\mathbf{B}_{\mathrm{dip},k} = -\frac{\delta F_{\mathrm{dip},k\neq 0}}{\delta \mathbf{M}}, \quad \text{for } k \neq 0, \tag{47}$$

from Eq. (45) to be added to $\mathbf{B}_{\mathrm{eff}}$ in the equation of motion Eq. (4). From this point on, it is just a matter of Taylor expanding $\mathbf{B}_{\mathrm{dip},k\neq 0}$ to first and second order in $\epsilon$ to add the relevant terms on the RHS of equations Eq. (6) and (7).

# D  Auxiliary calculations for excitations spectrum

## D.1  Equation of motion for $a, a^*$

In this section we show how the equation of motions of $a, a^*$ in Eq. (16) are obtained from Eq. (15). To calculate the excitation spectrum of the Archimedean screw solution, we first project Eq. (15) onto $\mathbf{e}_{\pm}$, defined in Eq. (14). We use the Holstein-Primakoff expansion Eq. (13) keeping only terms linear in $a, a^*$ on both sides of the equation. Note that the terms which are zeroth order in $a, a^*$ simply correspond to the LLG equation, which we solved correctly up to second order in amplitude of driving $b_x, b_y$ in Sec. 3. The following identities for the three basis vectors $\mathbf{e}_3, \mathbf{e}_{\pm}$ turn out to be useful

$$
\begin{aligned}
\mathbf{e}_3 \cdot \mathbf{e}_3 &= 1, & \mathbf{e}_{\pm} \cdot \mathbf{e}_{\pm} &= 0, \\
\mathbf{e}_{\pm} \cdot \mathbf{e}_{\mp} &= 1, & \mathbf{e}_{\pm} \cdot \dot{\mathbf{e}}_{\mp} &= \pm i \cos(\theta)\dot{\phi}, \\
\mathbf{e}_{\pm} \cdot \dot{\mathbf{e}}_{\pm} &= 0, & \mathbf{e}_{\pm} \cdot (\dot{\mathbf{e}}_3 \times \mathbf{e}_{\mp}) &= 0, \\
\mathbf{e}_{\pm} \cdot (\dot{\mathbf{e}}_{\pm} \times \mathbf{e}_3) &= 0, & \mathbf{e}_{\pm} \cdot (\dot{\mathbf{e}}_{\mp} \times \mathbf{e}_3) &= \cos(\theta)\dot{\phi}.
\end{aligned}
$$

Let us now project Eq. (15) onto $\mathbf{e}_+$. The terms linear in $a, a^*$ on the LHS are

$$\mathbf{e}_+ \cdot (\dot{\mathbf{e}}_- a + \mathbf{e}_- \dot{a} + \dot{\mathbf{e}}_+ a^* + \mathbf{e}_+ \dot{a}^*) = (i \cos(\theta)\dot{\phi} a + \dot{a}).$$

On the RHS of the projected Eq. (15) we have the Poisson bracket $\{F, \hat{\mathbf{M}}\}$. As we only consider $F^{(2)}$- the contribution to $F$ quadratic in $a, a^*$ - the only possibility for linear terms coming from the overall Poisson bracket is when we take the Poisson bracket with the linear in $a, a^*$ components $\mathbf{M}$, i.e. $\mathbf{e}_- a + \mathbf{e}_+ a^*$. After projecting onto $\mathbf{e}_+$ we obtain

$$\mathbf{e}_+ \cdot i\{F^{(2)}, \mathbf{e}_- a + \mathbf{e}_+ a^*\} = i\{F^{(2)}, a\}.$$

Next we consider the damping term $-\alpha \hat{\mathbf{M}} \times \dot{\mathbf{M}}$. Here there are two possibilities for obtaining linear terms in $a, a^*$: either we take a linear term from the first $\hat{\mathbf{M}}$ and zeroth term from $\dot{\mathbf{M}}$, or vice versa giving

$$\mathbf{e}_+ \cdot (\mathbf{e}_- a \times \dot{\mathbf{e}}_3 + \mathbf{e}_3 \times (\dot{\mathbf{e}}_- a + \mathbf{e}_- \dot{a} + \dot{\mathbf{e}}_+ a^* + \mathbf{e}_+ \dot{a}^*)) = i\dot{a} - \cos(\theta)\dot{\phi} a.$$

Setting the LHS equal to the RHS we obtain

$$\mathrm{sgn}(\gamma)(\dot{a} + i\cos(\theta)\dot{\phi} a) = i\{F^{(2)}, a\} - i\alpha(\dot{a} + i\cos(\theta)\dot{\phi} a).$$

Multiplying both sides of the equation by $\frac{\mathrm{sgn}(\gamma) - i\alpha}{1 + \alpha^2}$ we obtain the equation of motion for $a$ Eq. (16) given in the main text. The equation of motion for $a^*$ can be obtained by following the same procedure as above, but projecting onto $\mathbf{e}_-$ rather than $\mathbf{e}_+$, or simply by noticing that it should be the complex conjugate of Eq. (16).

## D.2 Derivation of Floquet matrix $M^F$

The goal of this subsection is to explain how we obtain the Floquet equation Eq. (18) from the equation of motion for the $a, a^*$ Eq. (16). The first step is to substitute the Fourier space and time expansions Eq. (17) into $a, a^*$ Eq. (16). The back Fourier transform lets us express $a, a^*$ as

$$
a(\mathbf{r}, t) = \sum_{\substack{k_\parallel, k_\perp \\ m, j \in \mathbb{Z}}} \tilde{a}^m_{j\mathbf{q}+\mathbf{k}} e^{-i\left(m\omega t + (jq+k_\parallel)(z+v_{\text{screw}}t) + \rho k_\perp\right)},
$$
$$
a^*(\mathbf{r}, t) = \sum_{\substack{k_\parallel, k_\perp \\ m, j \in \mathbb{Z}}} \tilde{a}^{-m*}_{-j\mathbf{q}-\mathbf{k}} e^{-i\left(m\omega t + (jq+k_\parallel)(z+v_{\text{screw}}t) + \rho k_\perp\right)},
$$
(48)

where $\mathbf{k} = \mathbf{k}_\parallel + \mathbf{k}_\perp$ and we are working in cylindrical coordinates $\mathbf{r} = (z, \boldsymbol{\rho})^T$, with $\boldsymbol{\rho} = (x, y)^T$. Due to the cylindrical symmetry of the problem, the azimuthal angle between $k_x, k_y$ makes no difference and can be set to 0. Also note that $k_\parallel$ is only defined in the first Brillouin zone, $-q/2 < k_\parallel < q/2$.

It is also useful to define the column vector $\Psi$ from which we the build the Floquet vector $\Psi^F$, Eq. (50).

$$
\Psi^m(\mathbf{k}) = \begin{pmatrix} \cdots & \tilde{a}^m_{\mathbf{k}-\mathbf{q}}, & \tilde{a}^{-m*}_{-\mathbf{k}-\mathbf{q}}, & \tilde{a}^m_{\mathbf{k}}, & \tilde{a}^{-m*}_{-\mathbf{k}}, & \tilde{a}^m_{\mathbf{k}+\mathbf{q}}, & \tilde{a}^{-m*}_{-\mathbf{k}+\mathbf{q}} & \cdots \end{pmatrix}^T,
$$
(49)

$$
\Psi^F(\mathbf{k}) = \begin{pmatrix} \cdots & \Psi^{-1}(\mathbf{k})e^{i\omega t}, & \Psi^0(\mathbf{k}), & \Psi^1(\mathbf{k})e^{-i\omega t}, & \cdots \end{pmatrix}^T.
$$
(50)

Substituting Eq. (48) into the LHS of Eq. (16) we obtain

$$
\dot{a} = \sum_{m,j \in \mathbb{Z}} (\dot{\tilde{a}}^m_{j\mathbf{q}+\mathbf{k}} - (im\omega + (jq + k_\parallel)v_{\text{screw}})\tilde{a}^m_{j\mathbf{q}+\mathbf{k}}) e^{-i\left(m\omega t + (jq+k_\parallel)(z+v_{\text{screw}}t) + \rho k_\perp\right)}
$$
(51)
$$
= \sum_{m,\text{odd } l} \left(\dot{\Psi}^m_l(\mathbf{k}) - i(m\omega + (f(l)q + k_\parallel)v_{\text{screw}}t)\Psi^m_l(\mathbf{k})\right) e^{-i(m\omega t + (f(l)q + k_\parallel)\tilde{z} + \rho k_\perp)},
$$

$$
\dot{a}^* = \sum_{m,j \in \mathbb{Z}} (\dot{\tilde{a}}^{-m*}_{-j\mathbf{q}-\mathbf{k}} - (im\omega + (jq + k_\parallel)v_{\text{screw}})\tilde{a}^{-m*}_{-j\mathbf{q}-\mathbf{k}}) e^{-i\left(m\omega t + (jq+k_\parallel)(z+v_{\text{screw}}t) + \rho k_\perp\right)}
$$
(52)
$$
= \sum_{m,\text{even } l} \left(\dot{\Psi}^m_l(\mathbf{k}) - i(m\omega + (f(l)q + k_\parallel)v_{\text{screw}}t)\Psi^m_l(\mathbf{k})\right) e^{-i(m\omega t + (f(l)q + k_\parallel)\tilde{z} + \rho k_\perp)},
$$

where we defined $\tilde{z} = z + v_{\text{screw}}t$ and $f(l) = \left\lfloor \frac{1}{2}(l - \frac{l_{\max}}{2}) \right\rfloor$, where $l$ runs between $l = 1$ and $l = l_{\max}$, and $l_{\max}$ stands for the maximal index of $\Psi$. It is sufficient to choose $l_{\max} = 6$ to obtain equations which are accurate to second order in the oscillating fields. $l_{\max}$ is always even because we always include the same number of $\tilde{a}^m_k$ and $\tilde{a}^{m*}_k$ operators. For the $\dot{a}$ expression we sum over only odd $l = 1, 3, \ldots l_{\max} - 1$, whereas for the $\dot{a}^*$ expression we sum over even $l = 2, 4, \ldots l_{\max}$.

Let's now look at the RHS of Eq. (16). First we have to compute the Poisson bracket $\{F^{(2)}, a/a^*\}$. As previously mentioned, $F^{(2)}$ is obtained by inserting Eq. (13) into Eq. (1) and keeping only the terms quadratic in $a, a^*$. By using the Fourier convention Eq. (17) we obtain $F^{(2)}$ in terms of the $\tilde{a}^m_{j\mathbf{q}+\mathbf{k}}, \tilde{a}^{m*}_{j\mathbf{q}+\mathbf{k}}$ operators. $F^{(2)}$ contains both number conserving operators $\tilde{a}^{m*}_{j\mathbf{q}+\mathbf{k}} \tilde{a}^n_{l\mathbf{q}+\mathbf{k}}$ and non-number conserving operators $\tilde{a}^m_{j\mathbf{q}+\mathbf{k}} \tilde{a}^n_{j\mathbf{q}+\mathbf{k}}, \tilde{a}^{m*}_{j\mathbf{q}+\mathbf{k}} \tilde{a}^{n*}_{j\mathbf{q}+\mathbf{k}}$. In general, this type of Hamiltonian can be diagonalized by Bogoliubov transformations, and the method we will use implicitly accomplishes the same thing. We denote the Fourier components of $F^{(2)}$ by $\tilde{F}^n(\mathbf{k})$. With this convention and the vectors $\Psi^m(\mathbf{k})$ defined in Eq. (49) we obtain

$$
F^{(2)} = \sum_{\mathbf{k}, n, m, l, j, j'} e^{-i\omega t(n-m+l)} \Psi^{m*}_j(\mathbf{k}) \tilde{F}^n(\mathbf{k})_{jj'}(\mathbf{k}) \Psi^l_{j'}(\mathbf{k}).
$$
(53)

Now, the Poisson bracket of $F^{(2)}$ with $a, a^*$ can be written in terms of the vector $\Psi_i^m$ as

$$
\begin{aligned}
\{F^{(2)}, a/a^*\} &= \sum_{\substack{\mathbf{k}, \mathbf{k}', j, j', j'' \\ n, m, l, m'}} e^{-i\omega t(n-m+l)} e^{-i(m'\omega t + (jq+k_\parallel)\tilde{z} + \rho k_\perp)} \tilde{F}_{j'j''}^n(\mathbf{k}') \left\{ \Psi_{j'}^{m*}(\mathbf{k}') \Psi_{j''}^l(\mathbf{k}'), \Psi_j^{m'}(\mathbf{k}) \right\} \\
&= \sum_{\mathbf{k}, n, l, j, j''} e^{-i\omega t(n+l)} e^{-i((f(j)q+k_\parallel)\tilde{z} + \rho k_\perp)} \\
&\qquad \times \left( (-1)^j \tilde{F}_{jj''}^n(\mathbf{k}) \Psi_{j''}^l(\mathbf{k}) + \left( \underbrace{\tilde{F}_{j',j+1}^n(-\mathbf{k})}_{j \text{ odd}} - \underbrace{\tilde{F}_{j',j-1}^n(-\mathbf{k})}_{j \text{ even}} \right) \Psi_{j'}^{-l*}(-\mathbf{k}) \right) \\
&= 2 \sum_{\mathbf{k}, n, l, j, j'} (-1)^j e^{-i\omega t(n+l)} e^{-i((f(j)q+k_\parallel)\tilde{z} + \rho k_\perp)} \tilde{F}_{jj'}^n(\mathbf{k}) \Psi_{j'}^l(\mathbf{k}),
\end{aligned}
\tag{54}
$$

where $j$ is odd if we are evaluating for the Poisson bracket with $a$ and even if it is the Poisson bracket with $a^*$. Above we used

$$
\begin{aligned}
\left\{ \Psi_i^m(\mathbf{k}), \Psi_j^{n*}(\mathbf{k}') \right\} &= (-1)^{i-1} \delta_{i,j} \delta_{m,n} \delta_{\mathbf{k}, \mathbf{k}'} \\
\left\{ \Psi_i^m(\mathbf{k}), \Psi_j^n(\mathbf{k}') \right\} &= (\delta_{i \in \text{odd}: i, j-1} - \delta_{i \in \text{even}: i, j+1}) \delta_{m,-n} \delta_{\mathbf{k}, -\mathbf{k}'} \\
\left\{ \Psi_i^{m*}(\mathbf{k}), \Psi_j^{n*}(\mathbf{k}') \right\} &= (\delta_{i \in \text{even}: i, j+1} - \delta_{i \in \text{odd}: i, j-1}) \delta_{m,-n} \delta_{\mathbf{k}, -\mathbf{k}'}
\end{aligned}
\tag{55}
$$

and

$$
\tilde{F}_{i,j}^n(\mathbf{k}) = \begin{cases}
\tilde{F}_{j+1,i+1}^n(-\mathbf{k}), & i \text{ odd}, \quad j \text{ odd} \\
\tilde{F}_{j-1,i+1}^n(-\mathbf{k}), & i \text{ odd}, \quad j \text{ even} \\
\tilde{F}_{j+1,i-1}^n(-\mathbf{k}), & i \text{ even}, \quad j \text{ odd} \\
\tilde{F}_{j-1,i-1}^n(-\mathbf{k}), & i \text{ even}, \quad j \text{ even}
\end{cases}
\tag{56}
$$

$$
\Psi_j^{-l*}(-\mathbf{k}) = \begin{cases}
\Psi_{j+1}^l(\mathbf{k}) & j \text{ odd} \\
\Psi_{j-1}^l(\mathbf{k}) & j \text{ even}
\end{cases}.
\tag{57}
$$

The final remaining term on the RHS of Eq. (16) is the $\dot{\phi} \cos(\theta)$ term. This term, which is built from the steady state solutions $\theta(z,t), \phi(z,t)$, oscillates in both space and time with components $e^{im\omega t}, e^{inq\tilde{z}}, m, n \in \mathbb{Z}$, and can be written as

$$
\dot{\phi} \cos(\theta) = \sum_{m,n \in \mathbb{Z}} e^{-i(m\omega t + nq\tilde{z})} g_n^m.
\tag{58}
$$

Finally, putting together Eq. (51), (54) and (58) and setting the coefficients of the terms which oscillate at the same spatial and temporal frequencies equal to each other, we obtain an equation of motion for $\Psi_j^m$

$$
\dot{\Psi}_j^m = i \left( (m\omega + (f(l)q + k_\parallel) v_{\text{screw}}) \delta_{m,l} \delta_{j,j'} + \frac{2 (\text{sgn}(\gamma)(-1)^j + i\alpha)}{1 + \alpha^2} \tilde{F}_{jj'}^{m-l} + (-1)^j g_{j-j'}^{m-l} \right) \Psi_{j'}^l.
\tag{59}
$$

From this, we can define the matrix $M^{ml}$ used to build the Floquet matrix $M^F$ with

$$M^{ml}_{jj'} = -\left((m\omega + (f(l)q + k_{\parallel})v_{\text{screw}})\delta_{m,l}\delta_{j,j'} + \frac{2\left(\text{sgn}(\gamma)(-1)^j + i\alpha\right)}{1+\alpha^2}\tilde{F}^{m-l}_{jj'} + (-1)^j g^{m-l}_{j-j'}\right),$$

(60)

$$M^F = \begin{pmatrix} \ddots & & & & \\ & M^{1,1} & M^{1,0} & M^{1,-1} & \\ & M^{0,1} & M^{0,0} & M^{0,-1}. & \\ & M^{-1,1} & M^{-1,0} & M^{-1,-1} & \\ & & & & \ddots \end{pmatrix}.$$

(61)

The Floquet matrix $M^F$ is non-Hermitian. Its complex eigenvalues describe the energy and decay rate of the the spin wave excitations on top of the Archimedean screw solution.

## E  Onset of Chaos

As already mentioned in the main text and visualized in Fig. 7, we find a transition to chaotic behavior at some critical strength of the driving field in our simulations, as can be expected from a driven nonlinear system with many degrees of freedom. In order to further examine this transition, we analyze the dynamics of a single spin in more detail. First, we determine $\Omega_{\text{screw}}$ from a linear fit to its azimuthal angle $\phi(t)$, see also Fig. 7. Then, we evaluate the orientation of the spin stroboscopically at times $t_n = 2\pi n/(\omega - \Omega_{\text{screw}})$, and rotate the result by an angle $-\Omega_{\text{screw}}t_n$ around $\mathbf{q} \parallel \mathbf{e}_z$, to eliminate the screw rotation. In Fig. 10 the projection of this spin onto the $xy$ plane is shown for a range of driving field amplitudes. For weak driving, $B_{\perp} \lesssim 0.6\,\text{mT}$ (panels (a) and (b)), we obtain (within the numerical accuracy) a single point, which is the signature of the Archimedean screw phase. For stronger driving, the time quasicrystal forms as discussed in Sec. 5. In this case the spin obtains an extra periodic oscillation, see also Fig. 7. Within our stroboscopic projection, this manifests in closed orbits visible in panels (c)–(o). For $B_{\perp} \gtrsim 4\,\text{mT}$, panel (p), chaos sets in. In Fig. 10, this manifests in aperiodic trajectories that fill certain regions of the plot. For stronger driving a larger area is filled, see panels (q)–(r).

We would like to emphasize that both the onset of chaos and the nature of the chaotic trajectories depends on the size of the unit cell used in our simulations (in Fig. 10 we use $15\frac{2\pi}{q}$). Smaller unit cells suppress chaos as they contain fewer degrees of freedom. Also in the chaotic regime, we expect translational invariance in the direction perpendicular to the helix not to be valid anymore.

## F  Transport calculation

In this section we calculate the current induced by the Archimedean screw solution in a metal. The task is to derive Eq. (33) using Keldysh diagrammatics. We consider a metallic disordered system and use the frame of reference where spins are locally rotated so that their spin-quantization axis aligns with the magnetization of the moving helix. As discussed in the main text, in this case the only time-dependent term arises from spin-orbit coupling of the electrons and is given by $H_1(t)$, see Eq. (30). In order to evaluate Eq. (32) up to second order

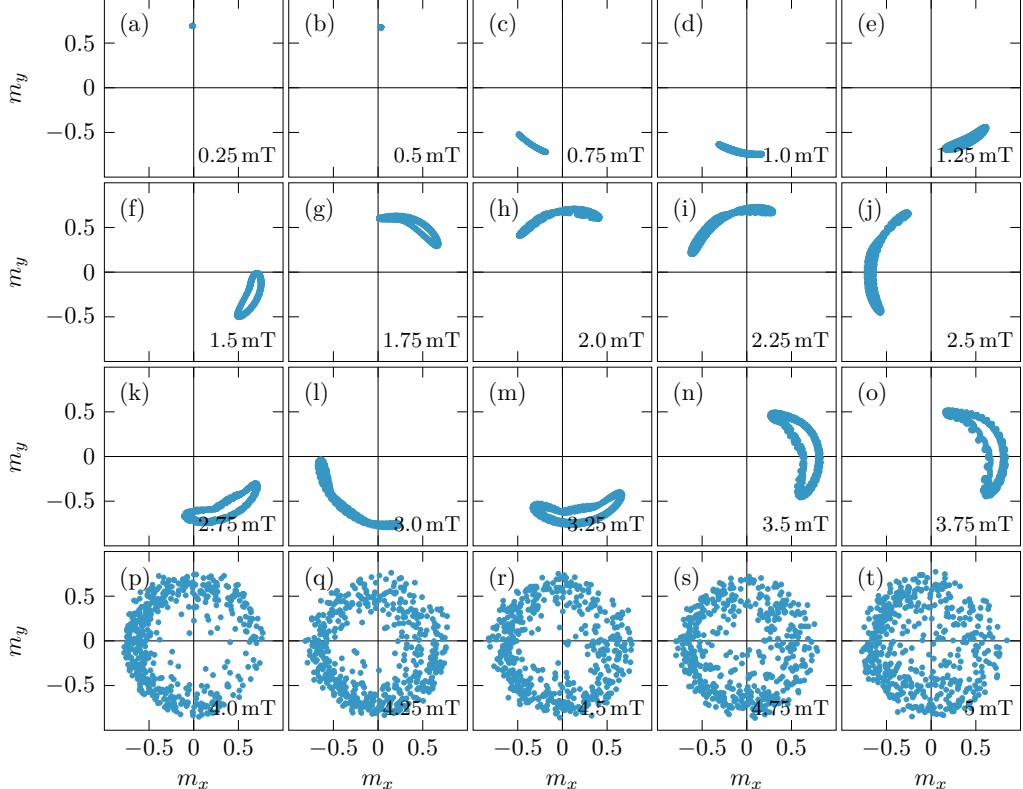

Figure 10: Projection of a single spin **m** onto the $xy$ plane recorded stroboscopically at times $t_n = 2\pi n/(\omega - \Omega_{\text{screw}})$, $n \in \mathbb{N}$, and rotated by $-\Omega_{\text{screw}} t_n$ to eliminate the screw rotation. In each panel, the respective amplitude $B_\perp^x$ is given in mT. Parameters are as in Fig. 8, for the system of size $15\frac{2\pi}{q}$. (a)–(b) In the regular regime, within numerical precision a single point is obtained. (c)–(o) A closed orbit signals the presence of the time quasicrystal. (p)–(r) The onset of chaos manifests itself in aperiodic trajectories, covering a significant area of the configuration space. Close to the onset of chaos we also see signatures of higher order time quasicrystals, with extra oscillation frequencies (panel (o)).

in $H_1(t)$, we need to expand the time-evolution operator $U(+\infty, -\infty)$ up to second order

$$U(+\infty, -\infty) \approx 1 - i \int_{-\infty}^{\infty} H_1(t)dt - \frac{1}{2}\iint_{-\infty}^{\infty} T[H_1(t)H_1(t')]dt\,dt'. \tag{62}$$

The expression for $U(-\infty, +\infty)$ is the same as Eq. (62) with the changes $-i \to i$, $T \to \tilde{T}$, where $\tilde{T}$ is the anti-time ordering operator.

Four different Green's functions of the free system are then needed to perform calculations on the Keldysh contour

$$G_\sigma^{++}(\mathbf{k},\omega) = \frac{-(1-n_{\sigma,\mathbf{k}})}{\omega - \epsilon_{\sigma,\mathbf{k}} - \frac{i}{2\tau}} - \frac{n_{\sigma,\mathbf{k}}}{\omega - \epsilon_{\sigma,\mathbf{k}} + \frac{i}{2\tau}},$$

$$G_\sigma^{--}(\mathbf{k},\omega) = \frac{1-n_{\sigma,\mathbf{k}}}{\omega - \epsilon_{\sigma,\mathbf{k}} + \frac{i}{2\tau}} + \frac{n_{\sigma,\mathbf{k}}}{\omega - \epsilon_{\sigma,\mathbf{k}} - \frac{i}{2\tau}},$$

$$G_\sigma^{+-}(\mathbf{k},\omega) = \frac{1-n_{\sigma,\mathbf{k}}}{\omega - \epsilon_{\sigma,\mathbf{k}} + \frac{i}{2\tau}} - \frac{1-n_{\sigma,\mathbf{k}}}{\omega - \epsilon_{\sigma,\mathbf{k}} - \frac{i}{2\tau}},$$

$$G_\sigma^{-+}(\mathbf{k},\omega) = \frac{n_{\sigma,\mathbf{k}}}{\omega - \epsilon_{\sigma,\mathbf{k}} - \frac{i}{2\tau}} - \frac{n_{\sigma,\mathbf{k}}}{\omega - \epsilon_{\sigma,\mathbf{k}} + \frac{i}{2\tau}}, \tag{63}$$

where we use the Fourier convention $G_\sigma(\mathbf{k},\omega) = \int dt\, e^{i\omega(t-t')} G_\sigma(\mathbf{k}, t-t')$ to switch between frequency and time domain. Here $n_{\sigma,\mathbf{k}} = (1 + e^{\beta(\epsilon_{\sigma,\mathbf{k}} - \epsilon_{\sigma,k_F})})^{-1}$ is the Fermi distribution function and $\epsilon_{\sigma\mathbf{k}}$ are the eigen-energies given in Eq. (29). We model the effects of disorder by a finite scattering rate $1/(2\tau)$. To simplify the calculation, we ignore vertex corrections arising from disorder as for short-ranged impurities they are expected to give only minor corrections.

We now have all the tools we need to evaluate Eq. (32). Using Wick's theorem, we obtain

$$\langle J_\parallel(t)\rangle \propto \iint_{-\infty}^{+\infty} dt_1 dt_2 \sum_{\sigma,\mathbf{k}_1,\mathbf{k}_2,\mathbf{k}} k_\perp^2 (k_\parallel - \sigma k_0) e^{-i\omega_{\text{screw}}(t_1-t_2)} \times$$

$$\langle T_C d_{\sigma,\mathbf{k}_1}^\dagger(t_1) d_{\sigma,\mathbf{k}_1+q}(t_1) d_{\sigma,\mathbf{k}_2+\mathbf{q}}^\dagger(t_2) d_{\sigma,\mathbf{k}_2}(t_2) d_{\sigma\mathbf{k}}^\dagger(t) d_{\sigma\mathbf{k}}(t)\rangle + h.c. \tag{64}$$

$$= \frac{1}{i} \sum_{\sigma,\mathbf{k}} k_\perp^2 (k_\parallel - k_{0,\sigma}) \iint_{-\infty}^{+\infty} dt_1 dt_2 e^{-i\omega_{\text{screw}}(t_1-t_2)} \times$$

$$\Big[ G_\sigma^{--}(\mathbf{k}, t-t_1) G_\sigma^{--}(\mathbf{k}, t_2-t) G_\sigma^{--}(\mathbf{k}+\mathbf{q}, t_1-t_2)$$

$$+ G_\sigma^{--}(\mathbf{k}, t_1-t) G_\sigma^{--}(\mathbf{k}, t-t_2) G_\sigma^{--}(\mathbf{k}-\mathbf{q}, t_2-t_1)$$

$$+ G_\sigma^{-+}(\mathbf{k}, t-t_1) G_\sigma^{+-}(\mathbf{k}, t_2-t) G_\sigma^{++}(\mathbf{k}+\mathbf{q}, t_1-t_2)$$

$$+ G_\sigma^{+-}(\mathbf{k}, t_1-t) G_\sigma^{-+}(\mathbf{k}, t-t_2) G_\sigma^{++}(\mathbf{k}-\mathbf{q}, t_2-t_1)$$

$$- G_\sigma^{-+}(\mathbf{k}, t-t_1) G_\sigma^{--}(\mathbf{k}, t_2-t) G_\sigma^{+-}(\mathbf{k}+\mathbf{q}, t_1-t_2)$$

$$- G_\sigma^{+-}(\mathbf{k}, t_1-t) G_\sigma^{--}(\mathbf{k}, t-t_2) G_\sigma^{-+}(\mathbf{k}-\mathbf{q}, t_2-t_1) + h.c. \Big]. \tag{65}$$

The next step consists of Fourier transforming the Green's functions in time, as well as time-averaging $\langle J_\parallel(t)\rangle$ to obtain the DC component $\langle J_{\parallel,DC}\rangle$. In addition, we can Taylor expand to first order in $\omega_{\text{screw}} = q v_{\text{screw}}$ (as $\omega_{\text{screw}}$ will be smaller than all electronic energy scales) to obtain

$$\langle J_\parallel\rangle \propto \frac{2q v_{\text{screw}}}{i} \int_{-\infty}^{\infty} \frac{d\omega}{2\pi} \sum_{\sigma,\mathbf{k}} k_\perp^2 (k_\parallel - k_{0,\sigma}) \times$$

$$\Big[ G_\sigma^{--}(\mathbf{k},\omega)^2 \partial_\omega G_\sigma^{--}(\mathbf{k}+\mathbf{q},\omega) + G_\sigma^{+-}(\mathbf{k},\omega) G_\sigma^{-+}(\mathbf{k},\omega) \partial_\omega G_\sigma^{++}(\mathbf{k}+\mathbf{q},\omega)$$

$$- G^{--}(\mathbf{k},\omega) \Big( G^{-+}(\mathbf{k},\omega) \partial_\omega G_\sigma^{+-}(\mathbf{k}+\mathbf{q},\omega) + G_\sigma^{+-}(\mathbf{k},\omega) \partial_\omega G_\sigma^{-+}(\mathbf{k}+\mathbf{q},\omega) \Big) \Big]. \tag{66}$$

Restoring prefactors and using cylindrical momentum coordinates, we obtain at $T = 0$

$$\langle J_{\parallel} \rangle = \sum_{\sigma=\uparrow,\downarrow} \tilde{J}_{\sigma} \int_{-k_{F,\sigma}}^{k_{F,\sigma}} \int_{k_{\perp}=0}^{\sqrt{k_{F,\sigma}^2 - k_{\parallel}^2}} \frac{dk_{\parallel} dk_{\perp} k_{\perp}^3 (q/2 - k_{\parallel})}{\left((k_{\parallel} - q/2)^2 + (q\tilde{\tau}^{-1})^2\right)^2},$$

$$\tilde{J}_{\sigma} = e N_{\sigma} v_{\text{screw}} \frac{3s^2 \lambda_{\text{so}}^2}{v_{F,\sigma}^3} \frac{\hbar}{qm}, \tag{67}$$

$$\tilde{\tau} = \frac{\hbar q^2 \tau}{m}, \ v_{F,\sigma} = \frac{\hbar k_{F,\sigma}}{m}.$$

Integrating first over $k_{\perp}$ and then by parts over $k_{\parallel}$ yields

$$\langle J_{\parallel} \rangle \simeq \sum_{\sigma=\uparrow,\downarrow} e N_{\sigma} v_{\text{screw}} \frac{3s^2 \lambda_{\text{so}}^2}{v_{F,\sigma}^3} \frac{\left(q^2 v_{F,\sigma}^2 \tau^2 + 3\right) \arctan\left(q v_{F,\sigma} \tau\right) - 3 q v_{F,\sigma} \tau}{2q\tau}, \tag{68}$$

where we have neglected small contributions of order $q/k_{F,\sigma}$. Taking the limits $v_F \tau \ll q^{-1}$, $v_F \tau \gg q^{-1}$ gives the result shown in Eq. (34).

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
