# Peer review of "Archimedean screw in driven chiral magnets"

_SciPost Physics, doi:SciPost Phys. 11, 009 (2021)_

## Round 1 · Referee Report · Anonymous (Referee 1) · 2021-4-24

Strengths

1 For the first time, this manuscript evaluated the basic dynamics and its stability of the driven chiral magnets, in particular the new mode, reminiscent of the motion of an Archimedean screw.
2 The analyses are multifold, from analytical study, numerical study as well as that from the Floquet spin-wave theory, which strengthen the reliability and clarify their applicability.
3 The authors had proposed ways to experimental verification of the phenomena, including the induced electron transport.

Weaknesses

1 As will be explained in the requested changes, the limiting condition of $\theta_0=\pi/2$ or equivalently $c=0$ must be treated properly.

Report

This manuscript contains detailed groundbreaking discovery on the properties of the Archimedean screw-type mode induced by an oscillating magnetic field and may stimulate a potential for multipronged follow-up experimental works.
The manuscript is written in a clear and intelligible way, free of unnecessary jargon and ambiguities. It contains a detailed abstract and introduction explaining the context of the problem and objectively summarizing the achievements and it provide sufficient details in appendices so that arguments and derivations can be reproduced by qualified experts. It also provides proper citations to relevant literature. I state the manuscript contains a clear conclusion summarizing the results with objective statements on their reach and limitations and offering perspectives for future work.

Requested changes

1 In page 5, after Eq. (2) and in the following arguments, the authors discuss the situation when $B_0=0$. Since the free energy $F$ prescribed in page 4 is isotropic when $\bf{B}_{\mathrm{ext}}$ is absent and is evatually spontaneous symmetry broken by the exchange interactions into an arbitrary magnetization axis. However, the authors had defined the $z$-axis as being parallel to $\bf{B}_0$, which makes no sense when $B_0=0$. Therefore, the authors need clearly explain the meaning of the magnetization vector for $\theta_0=0$, Eq. (2), for example as the limit of $B_0\to 0$.
2 Although the effect of the dipolar interaction is explained in detail in the Appendix, at least some related symbols in the main text should be explained. The undefined symbols are $N_x$ and $N_y$ appeared in page 6, the third paragraph.
3 In the main text and the caption to Fig. 4, the range of vertical axis of the upper two panels of Fig. 4 is in the first Floquet zone, $-\omega/2 < \mbox{Re}[\lambda]<\omega/2$ and $\lambda$ is said to be calculated modulo the driving frequency $\omega$. However, the range of the vertical axis is $[-1, 1]$. Why it is so ?
4 The energies of the spin waves $\epsilon_{i, k}^{0}$ had appeared in Eqs. (18) and (21), but in the text and in Eq. (20), they are instead $\epsilon_{i, k}$. The notation should be consistent.
5 The symbol $\lambda_{k}$ had appeared as the eigenvalues of the Floquet-Bogoliubov matrix. However, the symbol $\lambda$ is also appeared as the strength of the spin-orbit interaction of the conduction electrons in Eq.(24). To avoid the confusion, I suggest to use other symbol.

  • validity: high
  • significance: high
  • originality: high
  • clarity: top
  • formatting: excellent
  • grammar: good

Author:  Nina del Ser  on 2021-05-27  [id 1471]

(in reply to Report 1 on 2021-04-24)

We thank the referee for the very positive report and for giving suggestions to improve the paper. Below we cite the comments of the referee and provide a reply to all points raised.

1) “In page 5, after Eq. (2) and in the following arguments, the authors discuss the situation when $B_0=0$. Since the free energy F prescribed in page 4 is isotropic when $B_\text{ext}$ is absent and is evatually spontaneous symmetry broken by the exchange interactions into an arbitrary magnetization axis. However, the authors had defined the z-axis as being parallel to $B_0$, which makes no sense when $B_0=0$. Therefore, the authors need clearly explain the meaning of the magnetization vector for $\theta_0=0$, Eq. (2), for example as the limit of $B_0\to 0$.”

Both for finite and vanishing $B_0$ the oscillating field is perpendicular to the $\mathbf{q}$-vector describing the helical or conical state. We have modified the corresponding formulations in the text to clarify this issue.

2) “Although the effect of the dipolar interaction is explained in detail in the Appendix, at least some related symbols in the main text should be explained. The undefined symbols are $N_x$,$N_y$ appeared in page 6, the third paragraph.”

We explain in the revised version that the $N_i$ are the so-called demagnetization factors with more details provided by the appendices.

3) “In the main text and the caption to Fig. 4, the range of vertical axis of the upper two panels of Fig. 4 is in the first Floquet zone, $-\omega/2<\text{Re}[\lambda]<\omega/2$ is said to be calculated modulo the driving frequency omega. However, the range of the vertical axis is [-1,1]. Why it is so ?”

Thanks for pointing out the misleading caption used in the previous version. The plot is simply done for $w=2$ as we stress in the revised version of the figure caption.

4) “The energies of the spin waves $\epsilon^0_{i,k}$ had appeared in Eqs. (18) and (21), but in the text and in Eq. (20), they are instead $\epsilon_{i,k}$. The notation should be consistent.”

Thanks again, the notation in the revised version is hopefully consistent.

5 ) “The symbol $\lambda_k$ had appeared as the eigenvalues of the Floquet-Bogoliubov matrix. However, the symbol $\lambda$ is also appeared as the strength of the spin-orbit interaction of the conduction electrons in Eq.(24). To avoid the confusion, I suggest to use other symbol.”

We are following this suggestion and added the subscript "SO" when referring to the spin-orbit coupling.

---

## Round 1 · Referee Report · Anonymous (Referee 2) · 2021-4-27

Strengths

(1) The Archimedean screw function and the emergence of time quasicrystal predicted in this work are very innovative functionality and phenomenon of the driven conical magnetism in chiral magnets.
(2) A convincing analytical theory based on the Floquet formalism is provided.
(3) Numerical simulations are thoroughly performed to support the analytical theory.

Weaknesses

The manuscript contains many typos and errors.

Report

In this paper, the authors theoretically proposed an interesting dynamical phenomena of the conical magnetism in chiral magnets driven by a microwave magnetic field applied perpendicular to the conical propagation vector. They predicted the Archimedean-screw function carrying conduction electrons and formation of a time quasicrystal in the field-driven chiral magnets using both the analytical Floquet spin-wave theory and the numerical simulation with the LLG equation. The proposals and the predictions are quite attracting and novel, which elucidated new functionalities of helical and conical magnetisms in the dynamical regime. The formulation and the arguments are described in a convincing and persuasive manner in the paper. I think that this work will attract a great deal of research interest so that I recommend its publication in SciPost Physics. The followings are minor points, which may be helpful to improve clarity of the manuscript.

Requested changes

(1) Panels (a) and (b) in Fig.1 are better to be exchanged to be in an order that they appear in the main text.

(2) The usage of the terms ``helical" and ``conical" is confusing in the abstract and the introduction part. The readers may wonder if the conical nature, i.e., finite canting of magnetizations in the propagation direction is required for the predicted phenomena. If so, the authors may be better to avoid using the term ``helical".

(3) The experiment in Ref.[14] was done by the first two authors of this paper, Onose and Okamura, whereas Seki contributed to this work only by providing a sample of Cu2OSeO3. Thus, the sentence ``Early experiments by Seki et al [14] showed that ..." is not appropriate.

(4) The following sentence is rather difficult to understand:
``In the absence of pinning by disorder, this screw-like motion is induced for arbitrarily weak oscillating fields but requires some damping for its stability." If the authors intended to mean by this sentence that the resonant oscillation can be activated with an infinitesimally weak ac field in principle, but the oscillation amplitude becomes diverged in the absence of damping, this sentence may be better to be eliminated to avoid possible confusion because the damping or the dissipation is necessarily present in real systems, and the effect is usually assumed.

(5) The treatment of the sign of the gyrotropic ratio \gamma as a variable is rather strange to me and probably to other readers. It may be my prejudice, but usually \gamma is defined such that its sign is positive and the RHS of LLG equation reads
-\gamma M x B + (\alpha/M) M x \dot{M}. Anyway, I can understand that it depends. But even if the authors want to stick this treatment and definition, the definition of \gamma should be given in the text.

(6) The abbreviations RHS and LHS are used without definitions.

(7) A variable \lambda is used to denote both eigenvalues of the Floquet spin-wave Hamiltonian and strength of the spin-orbit interaction in Eq.(24), which may be a source of confusion. Maybe it is better to add a suffix ``so" to \lambda for the spin-orbit interaction.

(8) I found a lot of typos and minor errors in the text. For examples,
(8-1) In the second sentence below Eq.(12), one of the doubled ``where" should be eliminated.

(8-2) In the third sentence below Eq.(18), ``couples" should read ``couple".

(8-3) In the second sentence below Eq.(23), one of the doubled ``that" should be eliminated.

I also found that necessary commas are lacking at many points in the text. It may be better to use a commercial service to edit the manuscript.

(9) I wonder how the Floquet theory with the restricted number of ac drives to m=-1,0,1 is valid. Probably the validity is determined by the ratio of the magnon bandwidth (or the range of the eigenvalues \lambda) and the frequency of ac drives \omega. The authors focused on the eigenvalues in the m=0 subspace (the first Floquet zone), but there should be those in the m=+1 subspace above them and those in the m=-1 subspace below them with separations by \omega. If the frequency \omega is small as compared to the bandwidth, the bands in different subspaces should overlap. Moreover, if the frequency omega is much smaller, not only the |m|=1 subspace but also subspaces of higher |m| should affect the band structure in the first Floquet zone. I think that the authors' treatment with the restricted is validated in a certain range of frequency. I guess that this is a reason why the plots in Fig.6 are presented above \omega=1.5. I think that some comments on the present perturbative treatment will be helpful and informative for readers.

(10) To me, estimate of the induced electric current density argued in the last part of the manuscript is a little optimistic. The authors should, at least, mention references from which they adopt the parameter values of e.g., the strength of the spin-orbit interaction, the Fermi velocity, and the electron density. It is known that these parameter values vary depending on material species. Are they typical values for chiral magnets such as Cu2OSeO3 in Ref.[14]?

  • validity: high
  • significance: top
  • originality: top
  • clarity: high
  • formatting: excellent
  • grammar: good

Author:  Nina del Ser  on 2021-05-27  [id 1472]

(in reply to Report 2 on 2021-04-27)

We thank the referee for the very positive report and for giving suggestions to improve the paper. Below we cite all comments of the referee and provide a reply.

1) “Panels (a) and (b) in Fig.1 are better to be exchanged to be in an order that they appear in the main text.”

We followed this recommendation.

2) “The usage of the terms “helical" and “conical" is confusing in the abstract and the introduction part. The readers may wonder if the conical nature, i.e., finite canting of magnetizations in the propagation direction is required for the predicted phenomena. If so, the authors may be better to avoid using the term helical".

The conical nature (and thus an external magnetic field) is only required when one drives the system using linearly polarized oscillating fields. A rotation of the helical state can be induced by elliptically polarized fields. We have modified the introduction to make clear that rotations can be induced both for helical and conical states.

3) “The experiment in Ref.[14] was done by the first two authors of this paper, Onose and Okamura, whereas Seki contributed to this work only by providing a sample of $\text{Cu}_2\text{O}\text{SeO}_3$. Thus, the sentence Early experiments by Seki et al [14] showed that ..." is not appropriate.”

Thanks for pointing out this wrong citation, which has now been corrected.

4) “The following sentence is rather difficult to understand: In the absence of pinning by disorder, this screw-like motion is induced for arbitrarily weak oscillating fields but requires some damping for its stability." If the authors intended to mean by this sentence that the resonant oscillation can be activated with an infinitesimally weak ac field in principle, but the oscillation amplitude becomes diverged in the absence of damping, this sentence may be better to be eliminated to avoid possible confusion because the damping or the dissipation is necessarily present in real systems, and the effect is usually assumed.”

We follow the suggestion of the referee, omitting the stability discussion in this sentence.

5) “The treatment of the sign of the gyrotropic ratio $\gamma$ as a variable is rather strange to me and probably to other readers. It may be my prejudice, but usually $\gamma$ is defined such that its sign is positive and the RHS of LLG equation reads $-\gamma \mathbf{M} \times\mathbf{B} +\frac{ \alpha}{M} \mathbf{M} \times \dot{\mathbf{M}}$. Anyway, I can understand that it depends. But even if the authors want to stick this treatment and definition, the definition of $\gamma$ should be given in the text.”

Our impression is that different conventions for $\gamma$ are used (many textbook and, e.g., the wikipedia article on Larmor precession uses a convention with negative gamma). We found it useful for the analytic treatment to choose a convention where the sign of $\gamma$ can be chosen both positively and negatively. We changed the text below Eq. (4) to explain our convention.

6) “The abbreviations RHS and LHS are used without definitions.”

We corrected that.

7) “A variable $\lambda$ is used to denote both eigenvalues of the Floquet spin-wave Hamiltonian and strength of the spin-orbit interaction in Eq.(24), which may be a source of confusion. Maybe it is better to add a suffix "so" to $\lambda$ for the spin-orbit interaction.”

Thanks, we did that.

8) ”I found a lot of typos and minor errors in the text. For examples, 8-1) In the second sentence below Eq.(12), one of the doubled “where" should be eliminated. 8-2) In the third sentence below Eq.(18), “couples" should read couple. 8-3) In the second sentence below Eq.(23), one of the doubled “that" should be eliminated. I also found that necessary commas are lacking at many points in the text. It may be better to use a commercial service to edit the manuscript.”

Thanks again. We corrected the mentioned typos and eliminated a few other mistakes.

9) “I wonder how the Floquet theory with the restricted number of ac drives to $m=-1,0,1$ is valid. Probably the validity is determined by the ratio of the magnon bandwidth (or the range of the eigenvalues $\lambda$) and the frequency of ac drives $\omega$. The authors focused on the eigenvalues in the $m=0$ subspace (the first Floquet zone), but there should be those in the $m=+1$ subspace above them and those in the $m=-1$ subspace below them with separations by $\omega$. If the frequency \omega is small as compared to the bandwidth, the bands in different subspaces should overlap. Moreover, if the frequency omega is much smaller, not only the $|m|=1$ subspace but also subspaces of higher $|m|$ should affect the band structure in the first Floquet zone. I think that the authors' treatment with the restricted is validated in a certain range of frequency. I guess that this is a reason why the plots in Fig.6 are presented above $\omega=1.5$. I think that some comments on the present perturbative treatment will be helpful and informative for readers.”

Within our problem the use of a restricted number of Floquet modes is justified and controlled by the amplitude of the oscillating magnetic field. Our results for the Floquet spectrum are exact to second order in $B_{\perp}$. As we consider furthermore the limit of small alpha, the onset of the instability occurs in the regime of small $B_{\perp}$. This is the reason why our analytical approach can quantitatively predict the onset of the instability of the Archimedean screw with high precision. To emphasize this aspect, we added a corresponding remark to the text below Eq. (17).

The reason why in Fig. 6 only frequencies in the range $1.5-3.3$ are shown is actually a trivial one. In the figure we study the leading instability and for smaller frequencies this leading instability (obtained from Eq. (21)) is simply absent.

10) “To me, estimate of the induced electric current density argued in the last part of the manuscript is a little optimistic. The authors should, at least, mention references from which they adopt the parameter values of e.g., the strength of the spin-orbit interaction, the Fermi velocity, and the electron density. It is known that these parameter values vary depending on material species. Are they typical values for chiral magnets such as $\text{Cu}_2\text{OSeO}_3$ in Ref.[14]?”

After submission to SciPost, we discovered a mistake in the calculation of the current which we corrected. This leads to a reduced current in the dirty limit. Motivated by the question of the referee, we took more care in estimating parameters and realized that the previous order of magnitude estimate was too conservative and much too low when applied to a material like MnSi. To emphasize that our calculation can only provide a rough estimate we decided to give a range of values for the predicted current density instead of a single number. We predict giant current densities for materials like MnSi ranging from $10^4 - 10^7$ Am$^{-2}$. The revised version includes the new estimates and references for all relevant parameters.

---

## Round 2 · Referee Report · Anonymous · 2021-6-14

Report
I have checked all the changes which the authors made and found that all these minor revisions never affect the conclusion and discussion as well as the validity and importance of this work, while they improved the quality of presentations and arguments. I will not change my decision of strong recommendation for publishing this manuscript. I appreciate that the authors seriously considered and addressed the comments in my previous report.
Anonymous on 2021-06-23 [id 1515]
I have checked all the authors' replies and revised manuscript. The authors had revised the manuscript accordingly to all of the comments, which had improved the clarity and the strength of the arguments. I state that now the manuscript is worth to be published in this Journal.

---

## Round 2 · Author Response

List of changes
We have brought a number of changes to the paper, the most notable ones being the following:
1) Corrected equation numbering (previously Eq.(1)-the free energy of the system- was missing a number).
2) Fig.1 panels exchanged to better reflect the order in which we refer to them in the text.
3) Reference [14] “Seki et al.” on page 3, bottom paragraph corrected to “Onose et al.”
4) Page 4, paragraph 2, last sentence: removed “requires some damping for its stability” as damping is already a given in experimental systems.
5) Page 5, under Eq(3): clarified that z is the direction assumed by vector q as the direction of spontaneous symmetry breaking in the absence of an external magnetic field B0.
6) Page 5, under Eq.(4): gave a definition for gyromagnetic ratio gamma and explained our choice of sign convention.
7) Page 6, first new paragraph under Eq.(8): exchange the word order to make it clear that Nx, Ny are the demagnetization factors.
8) Page 10, Fig. 4: updated the caption to emphasize that for driving frequency ω=2, the first Floquet zone lies between -1<Re[ ω]<1.
9) Page 11, under Eq.(17): added the sentence “The restriction of the Floquet space is formally justified because we investigate the system in the limit of small B_⊥ and our results for eigenenergies and decay rates are formally exact to quadratic order in B_⊥." to justify why the restriction of Floquet copies to m=-1,0,1 is valid for our case. See also reply to referee 2 for further details.
10)Page 11&12, Eq.(19),(21) and text between Eq(21)-(22) added missing “0” superscript to unperturbed energies ϵ_(i,k)^0, and missing minus signs for negative momentum.
11) Pages 15,16 and 17: changed all spin-orbit coupling constants λ→λ_SO to avoid confusion with Floquet eigenenergies from the previous section.
12)Page 17, Figure (9) and caption as well as Eq.(34) and the accompanying text all corrected for a mistake in the calculation of the current in the dirty limit. In this limit, current actually depends quadratically on τ, instead of being independent as previously claimed. Eq. (68) in App. F from which these results are derived was also corrected.
13)Page 18, paragraph 3: motivated by referee comments, we calculated ω_screw and v_screw for MnSi (micromagnetic parameter, as well as driving field and damping stated with references provided in the text), obtaining 10 MHz and 200 mm/s respectively, a factor 10 larger than for CSO.
14)Page 18, paragraph 4: again, motivated by referee comments, using experimental order of magnitude estimates for the spin-orbit coupling, the mean free path of electrons and electron density in MnSI and arrived at a current density order-of-magnitude estimate of 10^4-10^7 A/m^2. References where these experimental values can be found were also provided in the text.

---

## Round 2 · List of Changes

We have brought a number of changes to the paper, the most notable ones being the following:
1) Corrected equation numbering (previously Eq.(1)-the free energy of the system- was missing a number).
2) Fig.1 panels exchanged to better reflect the order in which we refer to them in the text.
3) Reference [14] “Seki et al.” on page 3, bottom paragraph corrected to “Onose et al.”
4) Page 4, paragraph 2, last sentence: removed “requires some damping for its stability” as damping is already a given in experimental systems.
5) Page 5, under Eq(3): clarified that z is the direction assumed by vector q as the direction of spontaneous symmetry breaking in the absence of an external magnetic field B0.
6) Page 5, under Eq.(4): gave a definition for gyromagnetic ratio gamma and explained our choice of sign convention.
7) Page 6, first new paragraph under Eq.(8): exchange the word order to make it clear that Nx, Ny are the demagnetization factors.
8) Page 10, Fig. 4: updated the caption to emphasize that for driving frequency ω=2, the first Floquet zone lies between -1<Re[ ω]<1.
9) Page 11, under Eq.(17): added the sentence “The restriction of the Floquet space is formally justified because we investigate the system in the limit of small B_⊥ and our results for eigenenergies and decay rates are formally exact to quadratic order in B_⊥." to justify why the restriction of Floquet copies to m=-1,0,1 is valid for our case. See also reply to referee 2 for further details.
10)Page 11&12, Eq.(19),(21) and text between Eq(21)-(22) added missing “0” superscript to unperturbed energies ϵ_(i,k)^0, and missing minus signs for negative momentum.
11) Pages 15,16 and 17: changed all spin-orbit coupling constants λ→λ_SO to avoid confusion with Floquet eigenenergies from the previous section.
12)Page 17, Figure (9) and caption as well as Eq.(34) and the accompanying text all corrected for a mistake in the calculation of the current in the dirty limit. In this limit, current actually depends quadratically on τ, instead of being independent as previously claimed. Eq. (68) in App. F from which these results are derived was also corrected.
13)Page 18, paragraph 3: motivated by referee comments, we calculated ω_screw and v_screw for MnSi (micromagnetic parameter, as well as driving field and damping stated with references provided in the text), obtaining 10 MHz and 200 mm/s respectively, a factor 10 larger than for CSO.
14)Page 18, paragraph 4: again, motivated by referee comments, using experimental order of magnitude estimates for the spin-orbit coupling, the mean free path of electrons and electron density in MnSI and arrived at a current density order-of-magnitude estimate of 10^4-10^7 A/m^2. References where these experimental values can be found were also provided in the text.

---

## Editorial Decision

published